# ROS-induced R loops trigger a transcription-coupled but BRCA1/2-independent homologous recombination pathway through CSB

Yaqun Teng[1,2,3], Tribhuwan Yadav[4], Meihan Duan[1,2,3], Jun Tan[3], Yufei Xiang[5], Boya Gao[3], Jianquan Xu[6], Zhuobin Liang[7], Yang Liu [6], Satoshi Nakajima[2,3], Yi Shi[5], Arthur S. Levine[2,3], Lee Zou[4,8] & Li Lan[2,3,4,9]

Actively transcribed regions of the genome are protected by transcription-coupled DNA repair mechanisms, including transcription-coupled homologous recombination (TC-HR). Here we used reactive oxygen species (ROS) to induce and characterize TC-HR at a transcribed locus in human cells. As canonical HR, TC-HR requires RAD51. However, the localization of RAD51 to damage sites during TC-HR does not require BRCA1 and BRCA2, but relies on RAD52 and Cockayne Syndrome Protein B (CSB). During TC-HR, RAD52 is recruited by CSB through an acidic domain. CSB in turn is recruited by R loops, which are strongly induced by ROS in transcribed regions. Notably, CSB displays a strong affinity for DNA:RNA hybrids in vitro, suggesting that it is a sensor of ROS-induced R loops. Thus, TC-HR is triggered by R loops, initiated by CSB, and carried out by the CSB-RAD52-RAD51 axis, establishing a BRCA1/2-independent alternative HR pathway protecting the transcribed genome.

[1] School of Medicine, Tsinghua University, No.1 Tsinghua Yuan, Haidian District, Beijing 100084, China. [2] Department of Microbiology and Molecular Genetics, University of Pittsburgh School of Medicine, 450 Technology Drive, 523 Bridgeside Point II, Pittsburgh, PA 15219, USA. [3] UPMC Hillman Cancer Center, 5117 Centre Avenue, Pittsburgh, PA 15213, USA. [4] Massachusetts General Hospital Cancer Center, Harvard Medical School, Boston, MA 02129, USA. [5] Department of Cell Biology, University of Pittsburgh School of Medicine, 3500 Terrace Street, S362 Biomedical Science Tower South, Pittsburgh, PA 15213, USA. [6] Department of Medicine and Bioengineering, University of Pittsburgh, 5117 Centre Ave, Pittsburgh, PA 15232, USA. [7] Department of Molecular Biology and Biophysics, Yale Medical School, 333 Cedar Street, New Haven, CT 06520, USA. [8] Department of Pathology, Massachusetts General Hospital, Harvard Medical School, Boston, MA 02115, USA. [9] Department of Radiation Oncology, Massachusetts General Hospital, Harvard Medical School, Boston, MA 02129, USA. Correspondence and requests for materials should be addressed to L.L. (email: llan1@mgh.harvard.edu)

Reactive oxygen species (ROS) arise from both cellular metabolism and environmental insults, presenting a major threat to genomic stability that contributes to tumorigenesis and neurodegenerative diseases[1,2]. ROS induce multiple types of DNA lesions, including oxidized bases, DNA single-strand breaks (SSBs) and double-strand breaks (DSBs), which are removed by different DNA repair pathways[3]. ROS-induced DNA damage in transcriptionally active regions of the genome may be particularly deleterious to cells. For example, DNA damage-induced stalling of RNA polymerase II (RNAPII) may directly impair gene expression[4]. Furthermore, DNA damage in transcribed regions may lead to mutations, indels, and translocations in critical genes, driving tumorigenesis and neurodegeneration. Therefore, it is crucial to understand how cells protect the actively transcribed genome against ROS-induced DNA damage.

Recently, a growing body of evidence suggested that active genes are protected by transcription-coupled DNA repair mechanisms[5,6]. We and others showed that transcription-coupled homologous recombination (TC-HR) occurs in human and yeast cells and contributes to DSB repair in transcribed regions[7,8]. In contrast to the canonical HR, TC-HR functions in a transcription-dependent manner. Furthermore, the RNA transcript generated by transcription is required for TC-HR. Notably, we showed that ROS activated TC-HR at a transcriptionally active locus, thereby implicating TC-HR in the repair of ROS-induced DNA damage in transcribed regions. Despite these tantalizing features, TC-HR is still poorly understood as a pathway. In particular, whether and how the canonical HR and TC-HR pathways are differentially initiated and regulated remains elusive.

In this study, we used an inducible system to generate ROS at a transcriptionally active locus and characterized the TC-HR pathway. We found that TC-HR requires the RAD51 recombinase but, surprisingly, not the canonical HR proteins BRCA1 and BRCA2. The recruitment of RAD51 to sites of ROS-induced DNA damage is dependent on transcription, as well as Cockayne Syndrome Protein B (CSB) and RAD52 proteins. During TC-HR, RAD52 is recruited to sites of damage by CSB through an acidic domain (AD). The recruitment of CSB requires DNA:RNA hybrids, which are strongly induced by ROS in the transcribed region. In vitro, CSB directly and robustly binds to DNA:RNA hybrids, suggesting that it is a sensor of ROS-induced R loops in transcribed regions. Together, these results suggest that ROS-induced R loops in transcribed regions trigger TC-HR through the CSB-RAD52-RAD51 axis, revealing the framework of an alternative HR pathway that protects the transcribed genome against ROS-induced DNA damage.

## Results

**RAD52 but not BRCA1/BRCA2 recruits RAD51 in TC-HR.** To understand how cells protect the actively transcribed genome against ROS-induced DNA damage, we used KillerRed (KR), a light-excitable and ROS-releasing chromophore, to conditionally generate DNA damage at a genomic locus in U2OS Tet Response Element (TRE) cells (Fig. 1a)[9]. An array of the TRE was inserted next to the promoter of a reporter gene and integrated in the genome. A fusion of the transcription activator VP16 and KR (TA-KR) binds to the TRE array, marks the locus, and activates transcription locally. In contrast to TA-KR, a fusion of the Tet repressor and KR (tetR-KR) binds the TRE array but does not activate transcription. Upon light activation, both TA-KR and tetR-KR release ROS locally, inducing equivalent amounts of DNA damage marked by γH2AX at the locus in the presence and absence of transcription, respectively (Supplementary Fig. 1a)[9]. Following damage induction, Ku70 and Ku80 are immediately

recruited to KR sites, showing the efficient induction of DSBs by ROS (Supplementary Fig. 1b)[9].

In TA-KR-expressing cells, the key HR protein RAD51 was readily detected at the locus marked by TA-KR upon light activation (Fig. 1a). In contrast, RAD51 was not efficiently detected at the tetR-KR locus, suggesting that RAD51 is preferentially recruited to the damage site in a transcription-dependent manner. To test whether RAD51 is functionally important for repairing ROS-induced DNA damage, we monitored the clearance of γH2AX over time (Fig. 1b and Supplementary Fig. 1c). At 1 h after light activation, TA-KR induced similar levels of γH2AX in RAD51 knockdown cells and control cells. However, H2AX levels were significantly higher in RAD51 knockdown cells and control cells. Furthermore, γH2AX levels were significantly higher in RAD51 knockdown cells than in control cells after 36 h, suggesting that DSBs were not efficiently repaired in the absence of RAD51. These results demonstrate that ROS-induced DNA damage at a transcribed locus triggers RAD51-dependent TC-HR.

RAD51 not only functions in HR but also protects stalled replication forks[10]. Blocking DNA replication by Aphidicolin (APH)[11] before damage induction did not affect the formation of γH2AX and RAD51 foci at the TA-KR locus (Supplementary Fig. 1d), showing that DNA replication is not required for the induction of DSBs by ROS and the activation of TC-HR. To test whether DNA replication affects the function of TC-HR, we followed DNA repair at the TA-KR site in the absence or presence of APH. Although APH increased γH2AX throughout the genome (Supplementary Fig. 1e, upper panel)[12], it did not affect the clearance of γH2AX at the TA-KR site 24–36 h after damage induction (Supplementary Fig. 1e, lower panels), indicating that repair had occurred through one or more APH-refractory polymerases. Notably, RAD51 was required for the clearance of γH2AX at the TA-KR locus regardless of the presence or absence of APH (Supplementary Fig. 1e, lower panels). Together, these results suggest that the function of RAD51 in TC-HR is independent of DNA replication.

In the canonical HR pathway, BRCA1 and BRCA2 promote the localization of RAD51 to DSBs[13]. In a previous study, we showed that BRCA1 and BRCA2 are also recruited to sites of ROS-induced DNA damage[9]. Surprisingly, knockdown of BRCA1 and BRCA2 did not affect the recruitment of RAD51 to the locus marked by TA-KR (Fig. 1c and Supplementary Fig. 2a). We then turned our attention to RAD52, which promotes the localization of RAD51 to DSBs in BRCA1/2-deficient cells[14,15]. We created RAD52 knockout (KO) U2OS TRE cells (Supplementary Fig. 2b). In RAD52 KO and knockdown cells, RAD51 foci were diminished at the TA-KR marked locus (Fig. 1d and Supplementary Fig. 2c). The defect of RAD51 localization in RAD52 KO cells was suppressed by exogenous RAD52 (Fig. 1e). Compared with control cells, RAD52 KO cells were defective in γH2AX clearance and this defect was not exacerbated by RAD51 knockdown (Fig. 1f), suggesting that RAD52 and RAD51 function in the same pathway.

DNA endonucleases are widely used in DNA repair studies to generate DSBs at defined genomic loci[7,16–19]. Repair of the I-SceI-generated DSBs in the DR-GFP reporter requires BRCA1/2 but not RAD52[20]. Consistently, I-SceI-generated DSBs in the TRE array recruited RAD51 in a BRCA1/2-dependent but RAD52-independent manner (Supplementary Fig. 2d). Thus, ROS and I-SceI activate two distinct HR pathways (Fig. 1g), raising a question as to how ROS affect DSB repair in transcribed regions.

**The CSB-RAD52-RAD51 axis promotes TC-HR.** To understand how ROS affect DSB repair, we first investigated how

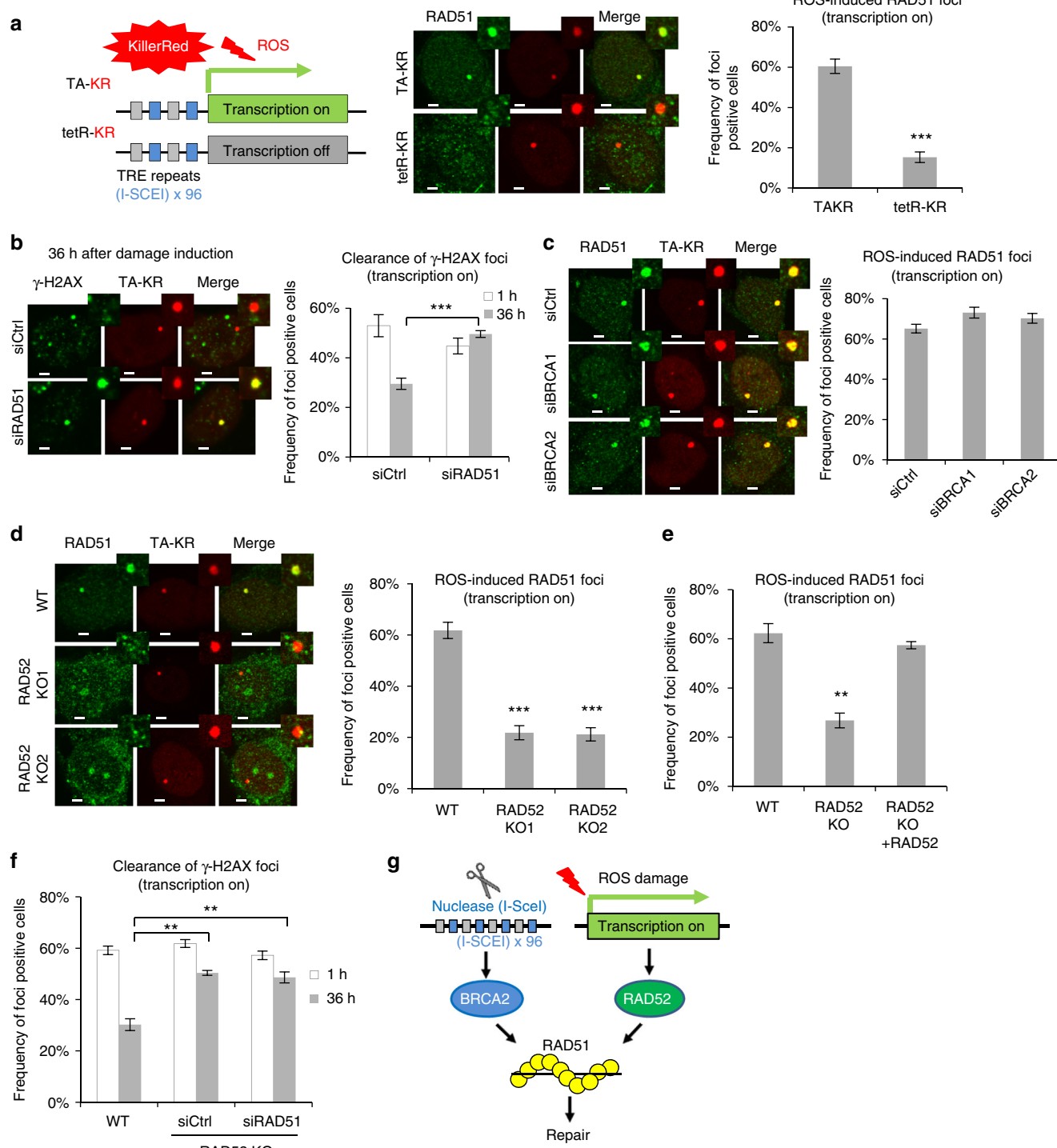

**Fig. 1** ROS trigger BRCA2/1-independent, RAD52-, and RAD51-dependent TC-HR. **a** Schematic diagram of the RAD51 damage response to KillerRed (KR)-mediated ROS-induced damage at transcription on (TA-KR) or off (tetR-KR) genomic loci in U2OS TRE cells (scale bar: 2 μm). **b** γH2AX foci frequency at TA-KR at early (1 h) or late (36 h) time points after light-induced KillerRed activation in siCtrl and siRAD51-treated cells (scale bar: 2 μm). **c** RAD51 foci frequency at TA-KR in cells treated with control, BRCA1, or BRCA2 siRNAs (scale bar: 2 μm). **d** RAD51 foci frequency at TA-KR in WT and RAD52 KO cells (scale bar: 2 μm). **e** RAD51 foci frequency at TA-KR in RAD52 KO cells and RAD52 overexpressed KO cells. **f** γH2AX foci frequency at TA-KR at early (1 h) and late (36 h) time points after damage in RAD52 KO cells treated with or without siRAD51. **g** Model of RAD51 regulation under ROS or nuclease-induced damage at transcribed regions. For **a**–**f**, $n = 3$, 50 cells per replicate. Unpaired $t$-test, error bars represent SEM. $*P < 0.05$, $**P < 0.01$, $***P < 0.001$

ROS-induced TC-HR is regulated. CSB is an important regulator of transcription-coupled nucleotide excision repair (TC-NER)[21]. We recently showed that CSB was recruited by ROS-induced DNA damage at a transcribed locus[9]. To understand the function of CSB in TC-HR, we created CSB KO U2OS TRE cells (Supplementary Fig. 3a). Similarly, in RAD52-depleted cells, the localization of RAD51 to the TA-KR-marked locus was diminished in CSB KO and knockdown cells (Fig. 2a–c and

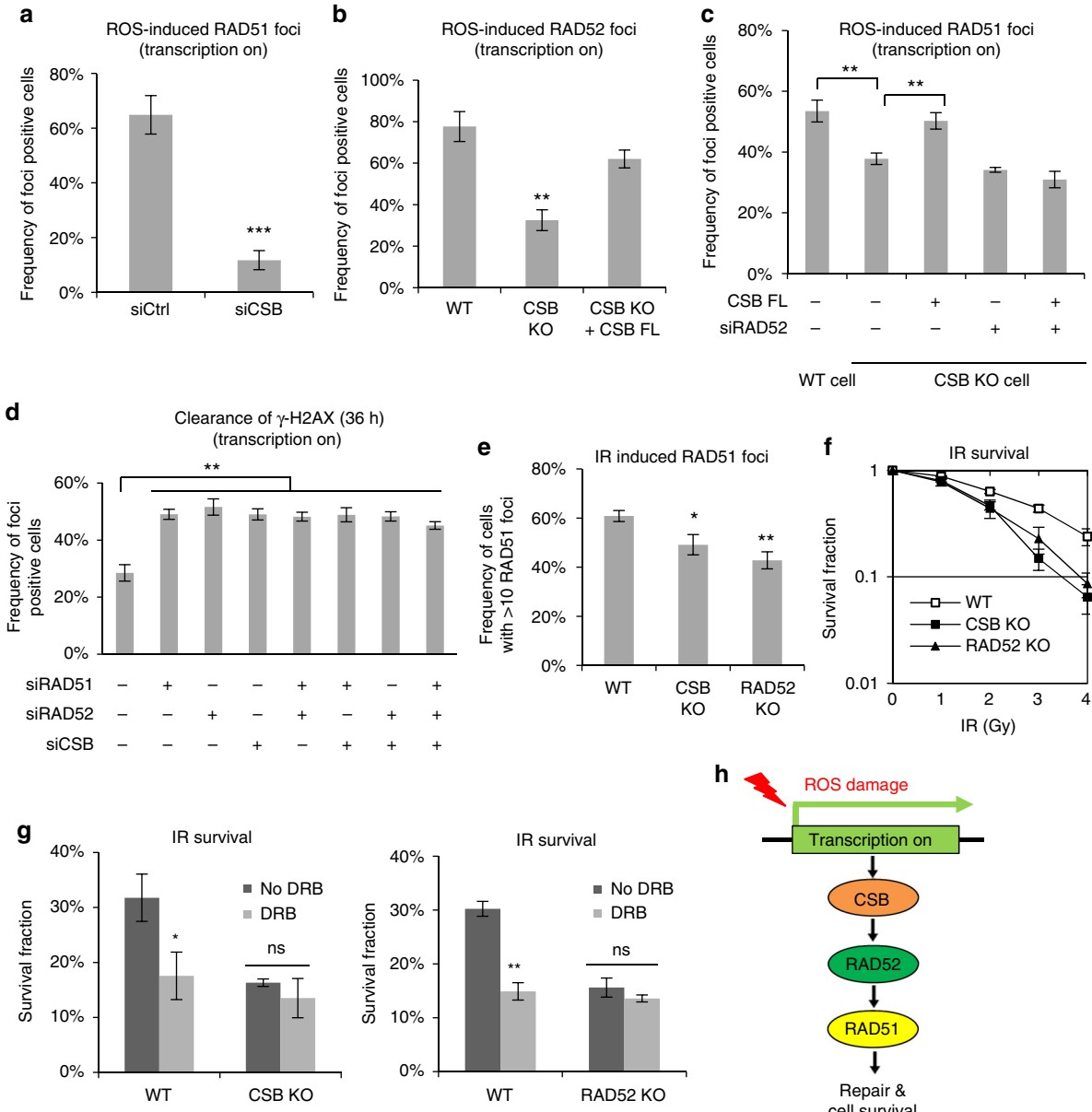

**Fig. 2** CSB promotes RAD51 recruitment to ROS-induced damage through RAD52. **a** RAD51 foci frequency at TA-KR in cells treated with control and CSB siRNAs. **b** RAD52 foci frequency at TA-KR in WT, CSB KO cells, and CSB KO cells transfected with full-length (FL) CSB. **c** RAD51 foci frequency at TA-KR in WT, CSB KO cells, and CSB full-length (FL) transiently transfected CSB KO cells with or without siRAD52 treatment. **d** γH2AX foci frequency at TA-KR at a late (36 h) time point after damage induction in cells with single, double, or triple knockdown of RAD51, RAD52, and CSB. For **a–d**, $n = 3$, 50 cells per replicate. **e** Frequency of cells (WT, CSB KO, and RAD52 KO) with more than 10 RAD51 foci per cell genome after 8 Gy ionizing radiation (IR) treatment and 6 h recovery ($n = 10$ views under the microscope were randomly selected and quantified in one experiment, 20–30 cells per view). **f** U2OS TRE WT, CSB KO, and RAD52 KO cell survival in a colony-formation assay under gradient IR treatment ($n = 3$). **g** Survival of WT, CSB KO, and RAD52 KO cells in a colony-formation assay under 3 Gy IR and additional DRB (20 μM, 24 h) treatment ($n = 3$). **h** Model of CSB-RAD52-RAD51 axis in response to ROS damage at transcription sites. Unpaired $t$-test, error bars represent SEM. $*P < 0.05$, $**P < 0.01$, $***P < 0.001$

Supplementary Fig. 3a-c), suggesting that CSB functions upstream of RAD51. As CSB is also required for the localization of RAD52 to the TA-KR-marked locus[9], CSB likely promotes RAD51 recruitment through RAD52. The defects of RAD52 and RAD51 localization in CSB KO cells were suppressed by exogenous full-length CSB (CSB-FL) (Fig. 2b-c). Notably, CSB-FL was unable to rescue RAD51 localization in the CSB KO cells treated with RAD52 small interfering RNA (siRNA) (Fig. 2c), confirming that CSB promotes RAD51 localization through RAD52. Knockdown of CSB, RAD52, and RAD51 either individually or in combinations resulted in similar defects in

knockdown of CSB (Fig. 2d and Supplementary Fig. 3d), suggesting that these proteins function in the same pathway. Knockdown of BRCA1/2 did not affect the localization of CSB and RAD52 to the TA-KR-marked locus (Supplementary Fig. 3e), supporting the notion that the CSB-RAD52-RAD51 axis functions independently of BRCA1/2.

Ionizing radiation (IR) induces ROS[22] and generates DSBs and other DNA lesions in both transcribed and untranscribed regions. We observed a mild defect in IR-induced RAD51 focus formation in CSB KO and RAD52 KO cells (Fig. 2e). Furthermore, CSB KO and RAD52 KO cells displayed

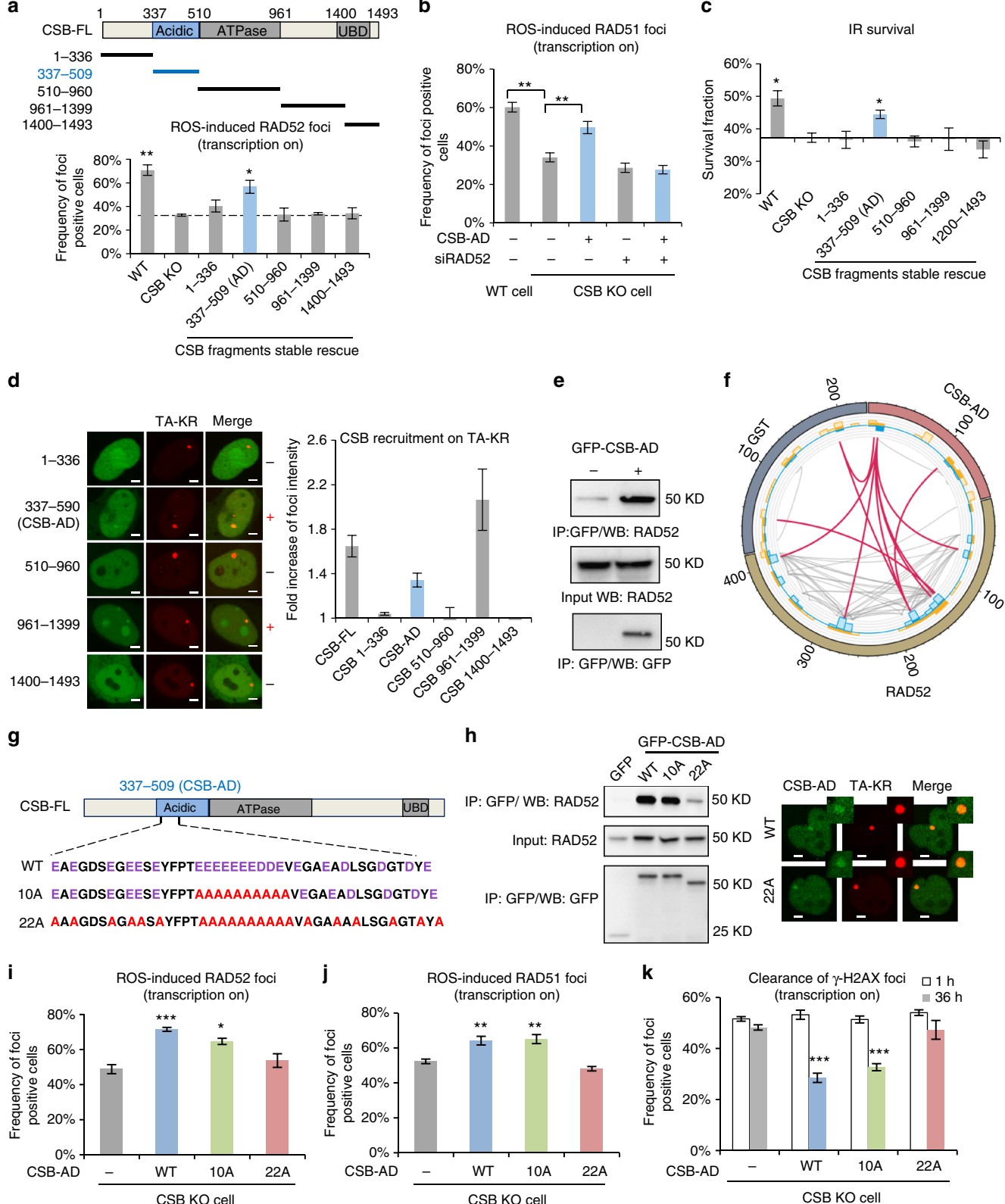

modestly increased IR sensitivity in the clonogenic assay (Fig. 2f), which was also observed in previous studies[14,15,23,24]. The RNAPII inhibitor 5,6-Dichlorobenzimidazole 1-β-D-ribofuranoside (DRB) sensitized BRCA2 knockdown cells to IR, but did not increase the IR sensitivity of CSB KO and RAD52 KO cells (Fig. 2g and Supplementary Fig. 3f). Therefore, transcription contributes to the repair of IR-induced

DNA damage independently of BRCA2 but in the same pathway as CSB/RAD52, which is consistent with the TC-HR pathway (Fig. 2h).

**CSB regulates RAD52 through an AD**. To understand how CSB regulates RAD52 in TC-HR, we stably expressed a panel

**Fig. 3** CSB recruits RAD52 via an acidic domain. **a** Schematic diagram of CSB fragments. GFP-RAD52 foci frequency is compared in U2OS TRE WT, CSB KO cells, and CSB fragment stable expression cells constructed by lenti-virus infection in CSB KO cells ($n = 3$, 50 cells per replicate). **b** RAD51 foci frequency at TA-KR in WT, CSB KO cells, and CSB-AD transiently transfected CSB KO cells with or without siRAD52 treatment ($n = 3$, 50 cells per replicate). **c** Survival of U2OS TRE WT, CSB KO and CSB fragment stable expression cell lines under 2 Gy IR in a colony-formation assay ($n = 3$). **d** The recruitment of GFP-CSB fragments to TA-KR sites. The CSB fragments showing colocalization with TA-KR are marked with +; the foci relative intensity was quantified ($n = 10$ cells in one experiment, scale bar: 2 μm). **e** Interaction of Myc-RAD52 and GFP-CSB-AD by anti-GFP Co-IP in Flp-in 293 cells. **f** Cross-linking between GST-CSB-AD and RAD52 protein analyzed by mass spectrometry. Red and gray lines indicate inter- and intra-molecule crosslinks, respectively. **g** Schematic diagram of CSB-AD10A and 22A. **h** Interactions between Myc-RAD52 and GFP-CSB-AD WT, 10A, and 22A were tested by anti-GFP Co-IP in Flp-in 293 cells. The relative foci intensity of GFP-tagged CSB-AD WT and the 22A mutant to TA-KR in CSB KO cells is quantified ($n = 10$ cells in one experiment, scale bar: 2 μm). **i** GFP-RAD52 and **j** RAD51 foci frequency at TA-KR in CSB KO cells transfected with Myc-tagged CSB-AD WT, 10A, and 22A ($n = 3$, 50 cells per replicate). **k** γH2AX foci frequency at TA-KR at early (1 h) and late (36 h) time points after damage induction in CSB KO cells transfected with CSB-AD WT, 10A, and 22A ($n = 3$, 50 cells per replicate). Unpaired $t$-test, error bars represent SEM. *$P < 0.05$, **$P < 0.01$, ***$P < 0.001$

of CSB fragments in CSB KO cells (Supplementary Fig. 4a)[25]. Among the CSB fragments, only the fragment containing the CSB-AD (337–509 a.a.) partially rescued the formation of RAD52 foci at the TA-KR-marked locus in CSB KO cells (Fig. 3a). Furthermore, CSB-AD also partially rescued the formation of RAD51 foci in CSB KO cells (Supplementary Fig. 4b). Similar to CSB-FL, CSB-AD rescued RAD51 foci in a RAD52-dependent manner (Fig. 3b). Consistent with the rescue of RAD52 and RAD51 foci, CSB-AD partially suppressed the IR sensitivity of CSB KO cells (Fig. 3c). CSB-AD was able to localize to the TA-KR-marked locus, although less efficiently than CSB-FL (Fig. 3d). Another C terminus fragment of CSB also localized to the TA-KR-marked locus, suggesting that CSB localization is regulated by a bipartite mechanism. Collectively, these results suggest that CSB may function in TC-HR through its AD. As CSB-AD is dispensable for TC-NER upon UV damage[26], the functions of CSB in TC-HR and TC-NER are likely distinct.

An interaction between CSB-AD and RAD52 was observed in cell extracts and in binding assays using purified proteins (Fig. 3e and Supplementary Fig. 4c-d). To understand how CSB-AD interacts with RAD52, we treated purified CSB-AD and RAD52 proteins with disuccinimidyl suberate (DSS) to crosslink lysine residues in close proximity (Supplementary Fig. 4e). Mass spectrometry (MS) analysis detected multiple crosslinks between RAD52 and the N terminal region of CSB-AD (Fig. 3f and Supplementary Fig. 4f). The N terminal region of CSB-AD is highly enriched for acidic amino acids. To test whether these acidic amino acids are important for RAD52 binding, we mutated 10 or 22 of them into alanines in CSB-AD, resulting in CSB-AD10A and CSB-AD22A mutants (Fig. 3g and Supplementary Fig. 5a). Although CSB-AD22A was able to localize to the TA-KR-marked locus, its binding to RAD52 was significantly reduced compared with CSB-AD and CSB-AD10A (Fig. 3h). Another CSB-AD mutant harboring 12 acidic amino acid mutations outside of the 10A region (CSB-AD12A) still interacted with RAD52 (Supplementary Fig. 5a-b), showing that the 22 acidic amino acids may be redundant for RAD52 interaction. Indeed, CSB-AD, CSB-AD10A, and CSB-AD12A could efficiently rescue RAD52 foci in CSB KO cells, whereas CSB-AD22A could not (Fig. 3i and Supplementary Fig. 5c). Furthermore, CSB-AD and CSB-AD10A, but not CSB-AD22A, rescued RAD51 foci and suppressed the defect of γH2AX clearance in CSB KO cells (Fig. 3j-k). Importantly, the CSB-FL mutant containing the 22A mutations (CSB-22A) also failed to rescue RAD52 foci and γH2AX resolution in CSB KO cells (Supplementary Fig. 5d-e). These results collectively suggest that the interaction of CSB with RAD52 through the acidic amino acids in CSB-AD is critical for TC-HR. Interestingly, CSB homologs in mice and Zebrafish, but not yeast, display an enrichment of acidic amino acids (Supplementary Fig. 5f), suggesting that the regulation of RAD52 has evolved in vertebrates.

**ROS induce R loops at a transcribed locus**. The transcription dependency of TC-HR prompted us to investigate whether RNA is involved in activating the CSB-RAD52-RAD51 axis. We previously showed that expression of RNase H1, which cleaves the RNA in DNA:RNA hybrids, inhibited TC-HR[9,27]. This result raised the possibility that R loops, which contain DNA:RNA hybrids and displaced single-stranded DNA, are induced by ROS[28]. Indeed, using a monoclonal antibody (S9.6) that specifically recognizes DNA:RNA hybrids, we detected R loops at the locus marked by TA-KR (Fig. 4a). The R loop signals at the TA-KR-marked locus were reduced by expression of RNase H1, but not the catalytically inactive mutant D210N (Fig. 4b)[29]. In contrast to TA-KR, tetR-KR did not induce R loops (Fig. 4a), suggesting that R loop formation is transcription-dependent. A fusion protein that activates transcription but does not induce ROS (TA-mCherry) and another fusion protein that lacks both transcription activation and ROS-releasing activities (tetR-mCherry) did not generate R loops (Fig. 4a), suggesting that both transcription and DNA damage are needed for R loop formation. Consistently, inhibition of RNAPII reduced the R loops at the TA-KR-marked locus (Fig. 4c). DNA–RNA immunoprecipitation (DRIP) confirmed that R loops were induced at the TA-KR locus in a damage- and transcription-dependent manner (Fig. 4d and Supplementary Fig. 6a). In addition, super-resolution imaging of the TA-KR-marked locus with stochastic optical reconstruction microscopy (STORM) confirmed that R loops were indeed in close proximity to TA-KR (Fig. 4e). These results demonstrate that R loops are robustly induced by ROS when TC-HR is activated. ROS induce both SSBs and DSBs. In vitro, SSBs in transcribed DNA are a potent inducer of R loops[30]. The DSBs generated by an endonuclease only modestly increased DNA:RNA hybrids in transcribed regions[31,32]. Together, these findings suggest that ROS may induce R loops by generating SSBs and DSBs in transcribed regions, providing a possible explanation for the unique ability of ROS to induce TC-HR efficiently.

**CSB binds R loops at damage sites through its C terminus**. We next asked how R loops contribute to TC-HR activation. As expected, both CSB and RAD52 colocalized with R loops at the TA-KR-marked locus (Fig. 5a). Importantly, the localization of CSB and RAD52 at the TA-KR-marked locus was reduced by RNase H1, but not the D210N mutant (Fig. 5b and Supplementary Fig. 6b), showing that CSB and RAD52 are recruited to the damage site in an R loop-dependent manner. The role of R loops in recruiting CSB and RAD52 suggests that R loops may be functionally important for TC-HR. Indeed, removal of R loops by RNase H1 expression increased the γH2AX signal at the locus marked by TA-KR (Fig. 5c), suggesting that TC-HR was compromised. These results suggest that R loops are required for activating the CSB-RAD52-RAD51 axis in TC-HR. Interestingly, a recent paper suggested that the HR function of BRCA1 is impaired by its interaction with the transcription elongation

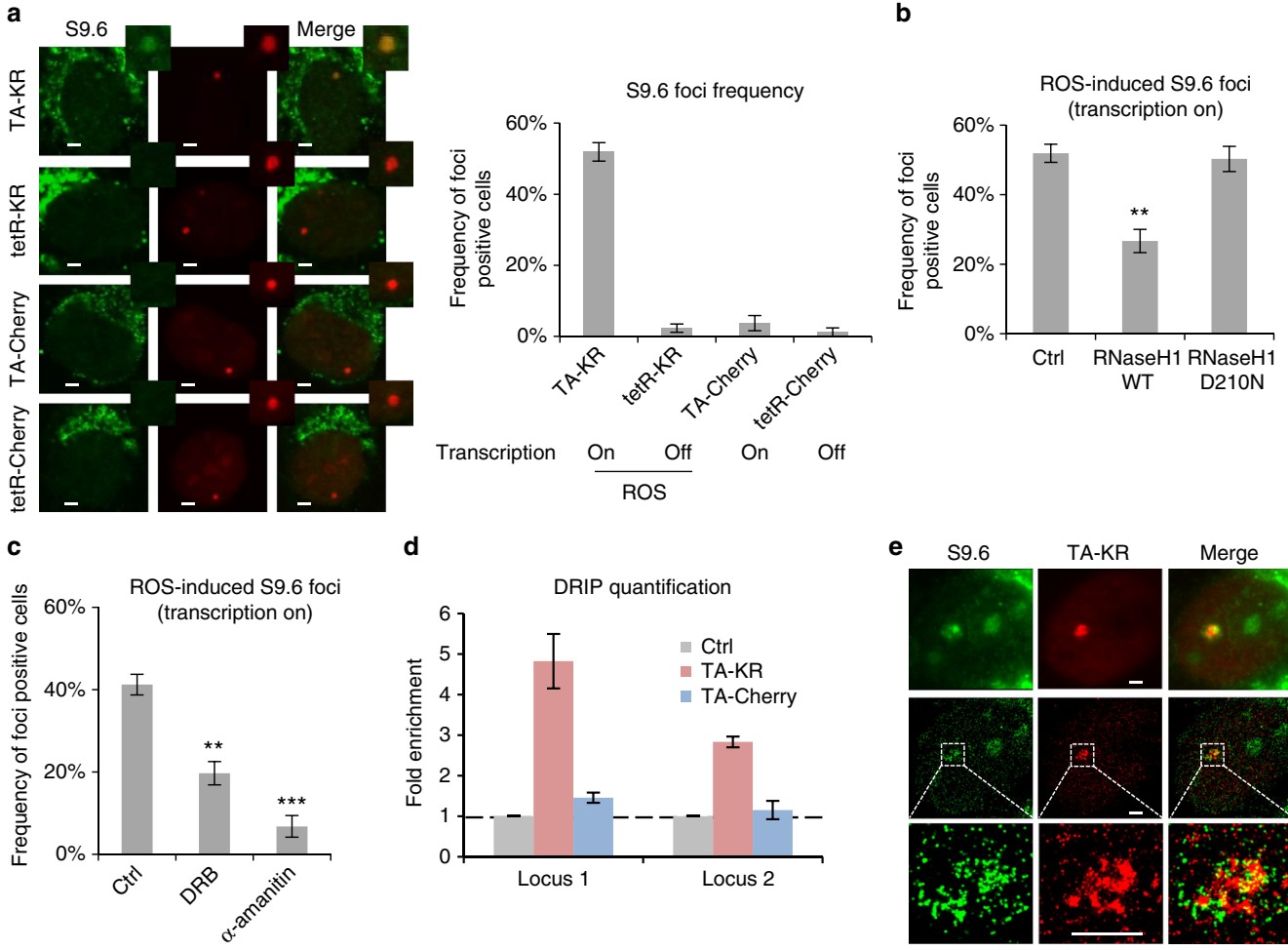

**Fig. 4** ROS induce R loops at transcribed regions. **a** S9.6 foci staining at TA-KR, tetR-KR, TA-Cherry, tetR-Cherry (scale bar: 2 μm). **b** S9.6 foci frequency at TA-KR in cells overexpressed with HA-RNase H1 WT and D210N mutant, or **c** treated with DRB (20 μM, 24 h) and α-amanitin (100 μg/ml, 2 h). For **a–c**, *n* = 3, 50 cells per replicate. **d** DRIP assay detected the fold enrichment of DNA:RNA hybrids at two loci in the transcription cassette near the TRE region in cells transfected with TA-KR/TA-Cherry (*n* = 3). **e** S9.6 co-staining with TA-KR visualized by super-resolution STORM (scale bar: 2 μm). Unpaired *t*-test, error bars represent SEM. *P < 0.05, **P < 0.01, ***P < 0.001

machinery in cells with high levels of R loops[33], raising the possibility that ROS-induced R loops may promote a switch from canonical HR to TC-HR in transcribed regions.

To address how R loops promote the recruitment of CSB and RAD52, we performed electrophoresis mobility shift assays using synthetic DNA:RNA hybrids and purified CSB-FL and RAD52 proteins (Supplementary Fig. 4c, 6c). Consistent with previous studies[9,34], RAD52 has an affinity for DNA:RNA hybrids (Fig. 5d). Strikingly, CSB-FL displayed a much higher affinity for DNA:RNA hybrids than RAD52 (Fig. 5d). These results suggest that both CSB and RAD52 may directly recognize the R loops at sites of DNA damage, and that CSB may enhance the recruitment of RAD52 to R loops by binding to both R loops and RAD52. The strong affinity of CSB to DNA:RNA hybrids in vitro, as well as its dependency on DNA:RNA hybrids to localize to sites of DNA damage in cells, suggest that CSB is likely a sensor of ROS-induced R loops that initiates the CSB-RAD52-RAD51 functional cascade.

To further understand how CSB senses R loops, we tested the binding of CSB fragments to DNA:RNA hybrids in cell extracts. Consistent with the bipartite mechanism of CSB localization to the TA-KR site, CSB-AD and the CSB C-terminal fragments 961–1493 and 1200–1493 were captured by DNA:RNA hybrids

(Supplementary Fig. 6d). Similar to CSB-AD, CSB 961–1493 and 1200–1493 were able to localize to the TA-KR site (Supplementary Fig. 6e). The CSB C-terminal fragments bound to DNA:RNA more efficiently than CSB-AD, suggesting that they contain the major DNA:RNA-binding domain of CSB. Importantly, using purified protein, we found that CSB 1200–1493 bound to DNA: RNA hybrids directly (Fig. 5e and Supplementary Fig. 6f), thereby revealing the DNA:RNA-binding domain of CSB. Unlike CSB 1200–1493, purified CSB-AD did not bind to DNA:RNA hybrids directly in vitro (Supplementary Fig. 6g), indicating that its localization to R loops might be mediated by other R loop-binding proteins.

## Discussion

Although the role of transcription in TC-NER has been long appreciated, its role in TC-HR has just begun to unfold. In this study, we used a unique experimental system to characterize ROS-induced TC-HR at a transcribed locus. We discovered that ROS induce R loops robustly in transcribed regions, recruiting CSB to sites of DNA damage through its C-terminal domain (CTD). Once recruited by R loops, CSB uses its AD to interact with RAD52 and promote its localization to DNA damage sites.

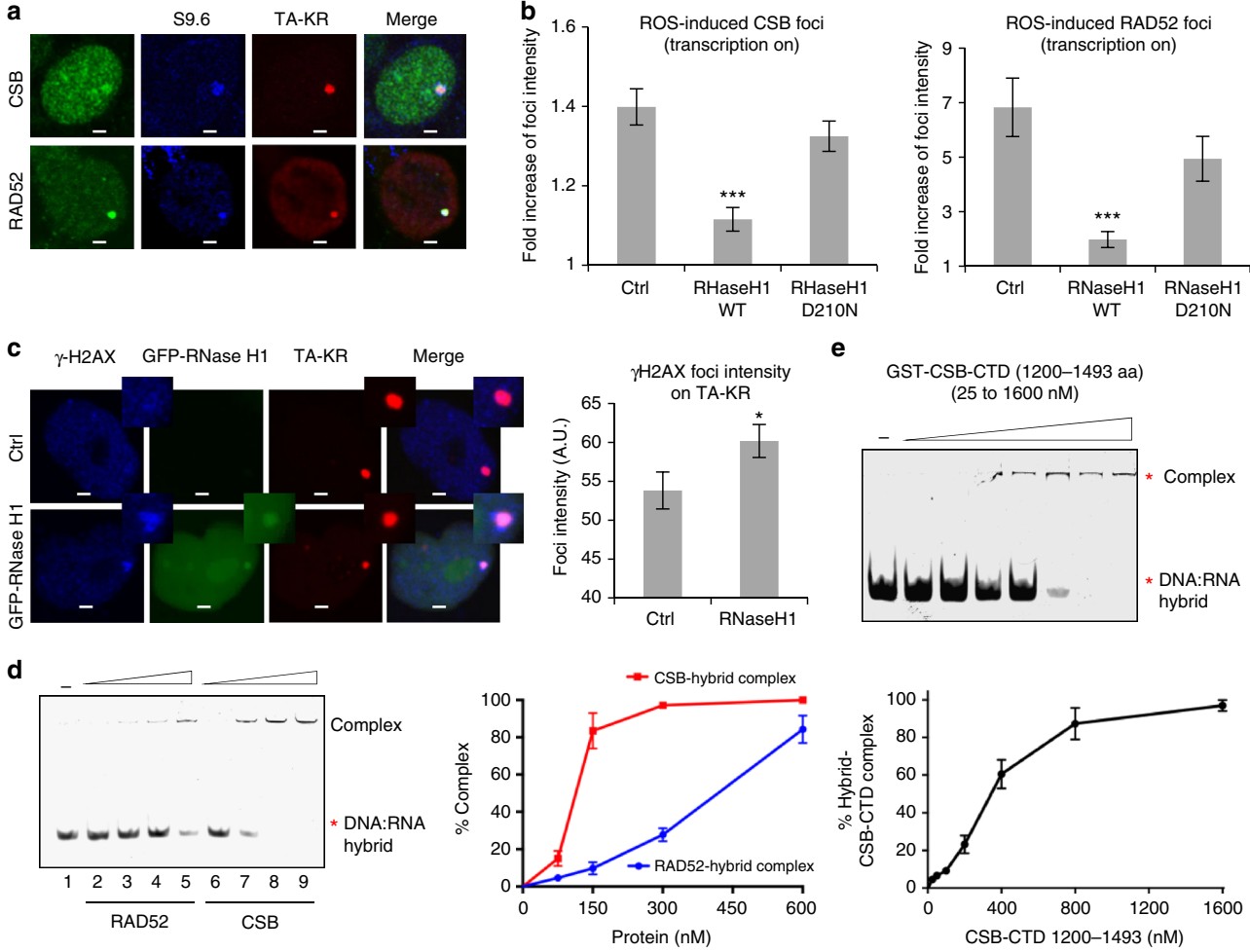

**Fig. 5** ROS-induced R loops activate TC-HR through CSB. **a** Colocalization of CSB or GFP-RAD52 with S9.6 foci at TA-KR (scale bar: 2 μm). **b** Myc-CSB or GFP-RAD52 foci relative intensity at TA-KR in cells overexpressed with HA-RNase H1 WT or D210N mutant (n = 10 cells in one experiment). **c** γH2AX foci (36 h) intensity at TA-KR in cells transfected with GFP-RNaseH after damage induction (n = 50 cells in one experiment, scale bar: 2 μm). For **b** and **c**, error bars represent SEM. **d** In vitro binding of CSB and RAD52 proteins to DNA:RNA hybrids in an electrophoretic mobility shift assay (EMSA). **e** DNA:RNA hybrids EMSA assay of purified GST-CSB-CTD (1200–1493) protein. For **d** and **e**, n = 3, error bars represent SD. Unpaired t-test, *P < 0.05, **P < 0.01, ***P < 0.001

The function of CSB in TC-HR is distinct from its function in TC-NER, revealing a new functional mode of CSB in transcription-coupled DNA repair.

Why do ROS preferentially activate TC-HR? Although high levels of R loops promote the recruitment of CSB and RAD52, they may inhibit the functions of certain canonical HR proteins. For example, in Ewing sarcoma, EWS-FLI1 induces R loops, increases the interaction between RNAPII and BRCA1, and inactivates BRCA1[33]. The high levels of R loops induced by ROS may also inhibit BRCA1. Through a mechanism that is still not fully understood, RAD52 promotes the localization of RAD51 to sites of DNA damage in transcribed regions in the absence of BRCA function. This is remarkably similar to the function of RAD52 in BRCA1/2-deficient cells[10,11]. Together, a functional cascade of CSB, RAD52, and RAD51 is triggered by ROS-induced R loops in transcribed regions, protecting the transcribed genome against ROS-induced DNA damage (Fig. 6a). As a BRCA1/2-independent alternative HR pathway triggered by ROS, TC-HR may be particularly important for suppressing ROS-induced genomic instability associated with tumorigenesis and degenerative neurological diseases. The TC-HR pathway may also function as a backup for canonical HR at DSBs generated in other contexts. With the key factors and events of TC-HR gradually defined and elucidated, we anticipate that more functions of TC-HR will be discovered in different biological contexts.

## Methods

**Cell culture, plasmids and siRNAs.** U2OS TRE, Flp-in 293 (Thermo), and 293 FT (ATCC) cells were cultured in Dulbecco's modified Eagle medium (DMEM, Lonza, Catalog#12-604F) with 10% (vol/vol) fetal bovine serum (FBS) at 37 °C, 5% $CO_2$. The U2OS TRE cell line for the DNA damage targeted at telomeres (DART) system is derived from wild-type U2OS cells (ATCC) by inserting an array of TRE/I-SceI and a transcription cassette in the genome[9]. pBROAD3/TA-KR, tetR-KR, TA-Cherry, tetR-Cherry, pCMV-NLS-I-SceI, pEGFP-RAD52[9], HA-RNaseH wild type, and HA-RNaseH D210N[27] plasmids were used for the DART system. CSB fragments 1–336, 337–509, 510–960, 961–1399, 961–1493, 1200–1493, and 1400–1493 were cloned into pEGFP-C1 and PLVX-IRES-Puro (Myc-tag) vectors using XhoI and NotI as digestion sites. The 510–960 fragment has an added NLS sequence in the N terminus to ensure nucleus localization. The 10A, 22A, and 12A mutants in the CSB 337–509 (CSB-AD) fragment were created using overlapping PCR strategy. The PCR primers for cloning are summarized in Supplementary Table 1. For purifying the GST-CSB-AD and GST-CSB 1200–1493 protein, the fragment CSB-AD and CSB 1200–1493 were inserted into a pGEX-4T-3 vector using XhoI and NotI digestion sites. Plasmids were transfected by Lipofectamine2000 (Invitrogen) using a standard protocol. siRNAs were transfected with Lipofectamine RNAiMax (Invitrogen) 48–72 h before analysis. The siRNAs used in this study are siCSB (SR320072, Origene), siRAD52 (GS5893, Qiagen), siRAD51 (E-003530–00, Dharmacon), siBRCA1 (L-003461-00, Dharmacon), and siBRCA2 (GS675, Qiagen).

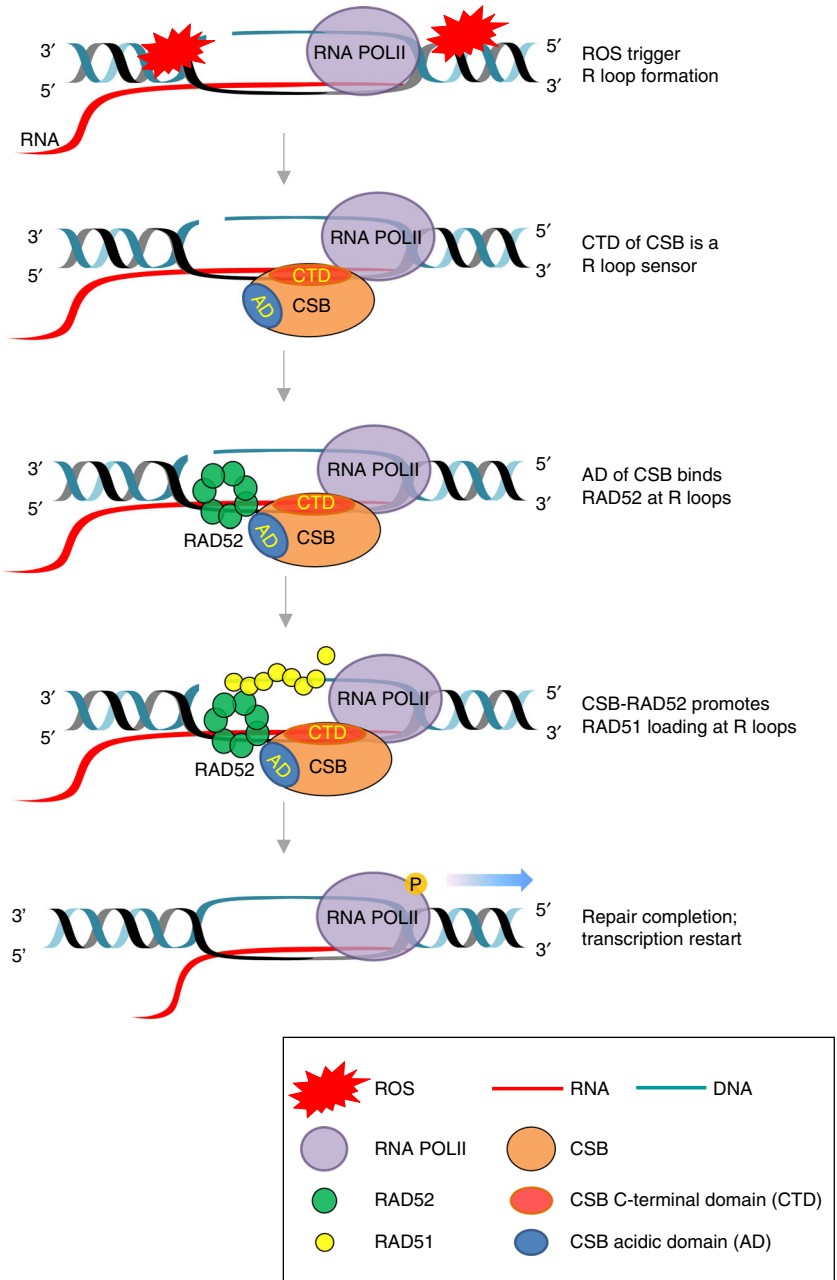

**Fig. 6** Model of CSB-, RAD52-, and RAD51-mediated TC-HR at ROS-induced R loops in transcribed regions. The current model is composed of the following sequential steps: ROS at transcribed regions trigger R loop formation; CSB senses R loops via its CTD; AD of CSB binds RAD52 and facilitates RAD52 recruitment at R loops; CSB-RAD52 promotes RAD51 loading at R loops; repair completion and transcription restart

**ROS and nuclease damage induction**. U2OS TRE cells were cultured in 35 mm glass-bottom dishes (MatTek, P35GC-1.5-14-C) at 60% confluence 24–36 h before the transfection. For ROS-induced damage, cells were transfected with plasmids containing KR (TA-KR/tetR-KR). Light-induced KR activation was done by exposing cells to a 15W Sylvania cool white fluorescent bulb for 25 min. Cells were recovered at around 1 h before live-cell observation or fixation. For γH2AX staining, cells were recovered for 36 h before fixation. For nuclease-induced damage, cells were co-transfected with pCMV-I-SceI plasmid and tetR-Cherry, and incubated for 36 h before harvest.

**Immunofluorescence staining and microscopy**. Cells in a 35 mm dish were rinsed with phosphate-buffered saline (PBS) and fixed in 4% paraformaldehyde (PFA; Affymetrix, 19943 1 LT) for 15 min at room temperature. They were washed three times by PBS, permeabilized by 0.2% Triton X-100 in PBS for 8 min, then washed three times by PBS. Then they were blocked by 5% bovine serum albumin (BSA) (SIGMA, A-7030) in 0.1% PBS-Tween (PBST) for 30 min at room temperature. Primary antibodies were diluted in blocking buffer and incubated with

cells overnight at 4 °C. Then the cells were washed three times with 0.05% PBST and incubated with secondary antibodies for 1 h at room temperature, including Alexa Fluor 405/488/594 goat anti-mouse/rabbit IgG conjugate (1: 10,000). Finally, they were washed three times by 0.05% PBST and optionally stained with DAPI (4′,6-diamidino-2-phenylindole; 1:1000 in PBS) for 5 min at room temperature. The primary antibodies for immunoassays are RAD51 (ab63801, Abcam, 1:100), γH2AX ser139 (JBW301, 05–636, EMD Millipore, 1:400), Myc-tag (ab9106 and ab32, Abcam, 1:200), HA (12CA5, 11666606001, Roche, 1:300), and S9.6 (ENH001, Kerafast, 1:200). APH was purchased from Abcam (ab142400).

For S9.6 staining, the cells were fixed and permeabilized in a 35 mm glass-bottom dish using a standard protocol, then incubated in buffer (10 mM Tris-HCl, 2 mM EDTA, pH = 9) and steamed on a 95 °C heating block for 20 min to expose the antigen. Then the dish was cooled, washed three times with PBS, and treated with RNase A (100 μg/mL) in buffer (5 mM EDTA, 300 mM NaCl, 10 mM Tris-HCl, pH 7.5) for 15 min at room temperature. Then the cells were washed and blocked using 5% BSA in 0.1% PBST for 0.5 h at room temperature. The first and secondary antibodies were diluted in the same buffer (5% BSA in 0.1% PBST) and the standard immunofluorescence protocol was followed. This protocol was

modified from the classical heat-induced antigen retrieval method for PFA-fixed tissues using Tris-EDTA buffer. For the RNAPII inhibitor, DRB (D1916; Sigma) or α-amanitin (A2263; Sigma) was added with a final concentration of 20 μM or 100 μg/mL for 24 h or 2 h before light irradiation, respectively.

The images were acquired using the Olympus FV1000 confocal microscopy system (Cat. F10PRDMYR-1, Olympus) and FV1000 software. For quantification of the percentage of foci-positive cells at sites of KR, 50 cells were counted in every experiment and representative data are shown. For quantification of relative foci intensity, the intensity of foci and background was acquired by ImageJ 1.50i software and the fold increase of foci is calculated as the foci intensity divided by background intensity. For quantification of RAD51 foci frequency after IR, the cells with more than ten RAD51 foci were counted and divided by the total cell number. The error bars represent SEM and the $P$-value was calculated by Student's $t$-test. $*P < 0.05$, $**P < 0.01$, $***P < 0.001$.

**Co-immunoprecipitation and western blots**. Flp-in 293 cells were co-transfected with expression vectors 36–48 h after transfection; cell lysates were collected in 1 mL of lysis buffer (10 mM Hepes, pH 7.6; 50 mM Tris-HCl, pH 7.4; 150 mM NaCl; 1% NP-40; 3 mM EDTA; protease inhibitor from Roche). For anti-green fluor-escent protein (GFP) immunoprecipitation, 2 μg anti-GFP monoclonal antibody (11814460001, Roche), and 25 μL of G-Sepharose protein beads (GE Healthcare Bio-Sciences) were added to each lysate. Mixtures were incubated at 4 °C overnight with rotation; the supernatant was removed and protein beads were washed four times using 0.4 mL of lysis buffer. For western blotting analysis, samples were boiled at 95 °C for 5–8 min in SDS loading buffer. Then they were subjected to electrophoresis in 10–12% SDS-polyacrylamide gels and transferred to the poly-vinylidene difluoride membrane. The membranes were blocked with 5% non-fat milk in PBS for 1 h before being incubated with primary antibody at 4 °C over-night. The primary antibodies for western blotting used in this study were GFP (11814460001, Roche, 1:2000), CSB (H-300, sc-25370, Santa Cruz Biotechnology, 1:400), RAD52 (F-7, sc-365341, Santa Cruz Biotechnology, 1:400), BRCA1 (D-9, sc-6954, Santa Cruz Biotechnology, 1:100), BRCA2 (3D12, sc-293185, Santa Cruz Biotechnology, 1:200), and β-Actin (8H10D10, Cell Signaling Technology, 1:10,000). Then the cells were washed three to four times with 0.1% PBST and incubated with horseradish peroxidase (HRP)-conjugated secondary antibody (1:10,000) for 1 h at room temperature. The membranes were washed in 0.1% PBST for four times before exposure. Chemiluminescent HRP substrate was pur-chased from Millipore (Catalog#: WBKLS0500). Images were acquired in a BIO-RAD Universal Hood II machine with ImageLab software. The uncropped scans of the important blots are displayed in Supplementary Fig. 7.

**DNA–RNA immunoprecipitation**. The DRIP assay was performed according to the literature with minor modifications[35]. U2OS TRE cells in a 10 cm dish were transfected with TA-KR or TA-Cherry, exposed to light for 20 min in PBS, and recovered for 1 h. The cells were washed and pelleted, resuspended in 1.6 ml of TE, and 41.5 μl of 20% SDS and 5 μl of Proteinase K (Roche Life Sciences) were added, and the cells were incubated overnight at 37 °C. The genomic DNA in DNA:RNA hybrids was extracted with phenol/chloroform in MaXtract High Density phase lock tubes (Qiagen), precipitated with EtOH/sodium acetate at room temperature gently without centrifugation, washed five times with 70% EtOH carefully, and resuspended in TE. DNA was digested with HindIII, BsrGI, XbaI, EcoRI, and SalI (the original SspI was replaced by SalI to avoid cutting the region of interest) at 37 °C overnight. DNA was purified by phenol/chloroform and precipitated with EtOH/sodium acetate, followed by 70% EtOH washes. Precipitated DNA (4.4 μg) was bound with 10 μg of S9.6 antibody in 1× binding buffer (10 mM NaPO4 pH 7.0, 140 mM NaCl, 0.05% Triton X-100) overnight at 4 °C. Protein G agarose beads were added for an additional 2 h. Antibody–DNA complexes were immunopreci-pitated and washed three times in 1× binding buffer, followed by 45 min incu-bation in Elution Buffer (50 mM Tris pH 8.0, 10 mM EDTA, 0.5% SDS, Proteinase K) at 55 °C with rotation. DNA was precipitated by EtOH/sodium acetate. The immunoprecipitated DNA fragments and input DNA was analyzed by PCR using primers: Locus 1F: 5′-TGTACGGTGGGAGGCCTATATAA-3′, Locus 1R: 5′-CGTCGCCGTCCAGCTCGACCAG-3′. Locus 2F: 5′-TTTCAAGGCAAT-CAGGGTAT-3′, Locus 2R: 5′-AGGCAGGATGATGACCAGG-3′′. The band intensity of PCR products was quantified by ImageJ and plotted.

**CRISPR-Cas9 KO**. The single-guide RNAs (sgRNAs) targeting CSB and RAD52 in the human genome were designed from the website http://crispr.mit.edu/ and cloned into PX330 vectors. The oligonucleotides for sgRNAs and PCR genotyping are summarized in Supplementary Table 2. The sgRNAs were delivered to cells by standard transfection. After 24 h, single cells were spread in 96-well plates or 10 cm dishes and grown for 10 days to obtain single colonies. The colonies were trans-ferred to 24-well plates and grown for about 1 week before genome extraction and genotyping, and western blotting verification. The uncropped scans of PCR gen-otyping and western blotting gels of CRISPR KO cell verifications are displayed in Supplementary Fig. 7.

**CSB fragment stable cell construction**. The Myc-tagged CSB fragments in PLVX-IRES-Puro vectors were co-transfected with packaging plasmids into 293

FT cells for virus packaging. Culture medium was changed 8 h after transfection. Forty-eight hours later, the medium was collected and filtered by a 0.45 μm filter (Millex HA, SLHAM33SS). Then U2OS TRE CSB KO cells were cultured in the medium and mixed with normal DMEM (10% FBS) at a 1:1 ratio. Polybrene (10 μg/mL) was added to the culture system to promote efficiency. Forty-eight hours later, the cells were cultured in DMEM (10% FBS) with 1 μg/mL puromycin and the medium was changed once every 2 days.

**Cell survival (colony-formation assay)**. Four hundred U2OS TRE cells were seeded in a 6 cm dish with or without DRB (20 μM). Eight hours after seeding, cells were exposed to IR. Twenty-four hours after seeding, the media containing DRB were replaced by fresh media. The cells were cultured for 7–9 days. Colonies were fixed and stained with 0.3% crystal violet in methanol and then the number of colonies was counted.

**Cell cycle analysis by flow cytometry**. The cells were collected and fixed in cold 70% ethanol at 4 °C overnight. The cells were washed once with 2% BSA in PBS and incubated in PBS solution containing 2% BSA, 50 μg/mL propidium iodide, and 100 μg/mL RNase A in the dark for 30 min before being analyzed by flow cytometry.

**Protein purification**. 6H-Flag-CSB was purified from High Five insect cells infected with baculovirus generated by the Bac-to-Bac® Baculovirus Expression System (Thermo Fisher) following the manufacturer's manual. All of the pur-ification steps were carried out at 4 °C. After 46 h of baculovirus infection, 800 ml of insect cells were lysed in Buffer A [25 mM Tris-Cl pH 7.5, 10% Glycerol, 0.5 mM EDTA, 300 mM KCl, 0.1% IGEPAL, 1 mM dithiothreitol (DTT), 1 mM pheyl-methylsulfonyl fluoride (PMSF), and protease inhibitors (5 μg/ml each of leupeptin, chymotrypsinogen, aprotinin, and pepstatin)] and sonicated for 30 s twice on a Bransen 250 sonifier set to power 4.5, 50% output. The lysate was subjected to ultracentrifugation for 60 min at 100,000 × g. The supernatant was added to 3 ml of Anti-flag M2 Affinity Agarose Gel (Sigma) pre-washed in Buffer A and rotated for 2 h. Beads were then washed four times, each with with 20 ml Buffer B (25 mM Tris-Cl pH 7.5, 10% Glycerol, 0.5 mM EDTA, 500 mM KCl, 0.01% IGEPAL, 1 mM DTT), followed by elution in 15 ml Buffer B with 200 μg/ml Flag peptide. The eluate was added to 3 ml of Nickel-NTA beads (Qiagen), pre-washed in Buffer B with 20 mM imidazole, and rotated for 2 h. Nickel-NTA beads were washed four times each with 10 ml Buffer B and 20 mM imidazole, followed by elution in 10 ml Buffer B with 200 mM imidazole. The eluate was subjected to size exclusion chromatography on a Superdex 200 column in Buffer B. Fractions containing CSB dimer species (~350 kDa) were pooled, concentrated using 100 kDa cutoff con-centrator, and stored in small aliquots at − 80 °C.

Rosetta cells harboring pET28b-RAD52 that encode hRAD52 with a C-terminal 6 × -his tag were grown to an optical density of 0.6 and induced by adding 0.3 mM Isopropyl β-D-1-thiogalactopyranoside (IPTG) for 3 h at 30 °C. Ten grams of overexpressed cell mass were lysed in lysis buffer [25 mM Tris-HCL (pH 7.5), 500 mM KCl, 1 mM EDTA, 10 % glycerol, 1 mM DTT, 0.01% Igepal, 1 mM PMSF, and a mixture of protease inhibitors] and sonicated. The lysed sample was centrifuged for 1 h at 16,000 r.p.m. (about 60,000 × g). Cleared supernatant was diluted five times and loaded on Affiblue beads in T buffer [25 mM Tris-HCl (pH 7.5), 1 mM EDTA, 10 % glycerol, 1 mM DTT, 0.01% Igepal] with 100 mM KCl. Using fast protein liquid chromatography, RAD52 protein was eluted by a gradient of 0–2.5 M of NaSCN in T buffer. Fractions containing RAD52 protein were pooled together and dialyzed against T buffer with 300 mM KCl, and then incubated with Ni-NTA agarose beads for 2 h. RAD52 protein was eluted by a gradient of 10–300 mM imidazole in T buffer. Fractions containing RAD52 protein were pooled together and dialyzed against T buffer with 300 mM KCl. Dialyzed RAD52 was concentrated and stored in − 80 °C for biochemical assays.

Glutathione S-transferase (GST)-tagged human CSB-AD was expressed in Rosetta 2 (Novagen) and purified with a glutathione sepharose 4B (GE Healthcare Life Sciences) column. Escherichia coli Rosetta 2 transformed with a plasmid expressing the GST-tagged human CSB-AD was grown at 30 °C in 1 L lysis buffer medium with 100 mg ampicillin and 15 mg chloramphenicol overnight. IPTG (final concentration; 0.1 mM) was then added to induce expression and the culture incubated for 4 h at 30 °C. Cells were collected, resuspended in extraction buffer (50 mM Tris-HCl pH 7.5, 0.1 M NaCl, 1 mM DTT, and 1% Triton X-100) and sonicated. The cell debris was removed by centrifugation. The cell-free extract was loaded onto a glutathione sepharose 4B (GE Healthcare Life Sciences) column.

CSB CTD (1200–1493 aa) was cloned in a pGEX-4T vector, transformed in Rosetta cells, and overexpressed with 0.3 mM IPTG at 160 °C for overnight. Two grams of cell mass was lysed in buffer 50 mM Tris-HCl (pH 7.5), 10% sucrose, 2 mM DTT, 0.01% NP-40, protease inhibitors cocktail, and 500 mM KCl, and centrifuged. Lysis supernatant was allowed to bind with Glutathione-Sepharose 4 beads for 2 h, washed, and protein was eluted with T buffer (50 mM Tris-HCl, 1 mM DTT, 0.01% NP-40, and 10 % glycerol) containing 100 mM KCl and 25 mM glutathione. Eluted fractions were dialyzed against T buffer containing 300 mM KCl, concentrated, and stored at − 80 °C.

**Hybrid substrate preparation**. The 5′-end of the oligo 1 (Supplementary Table 3) was labeled using a 5′-oligonucleotide end labeling kit (Vectorlabs) and a maleimide-IR800 probe (LI_COR Bioscience). RNA–DNA substrate was prepared by annealing oligo 1 and 2 (Supplementary Table 3) and confirmed[27]. Briefly, 5′-end-labeled oligo 1 was mixed with oligo 2 in buffer H [90 mM Tris-HCl (pH 7.5), 10 mM MgCl$_2$, and 50 mM NaCl], heat denatured, and annealed by slow cooling. Annealed substrates were separated by 10% native polyacrylamide gel electrophoresis (PAGE)-Tris-acetate-EDTA. The corresponding gel bands were excised and eluted. RNA–DNA hybrid substrate was confirmed by mobility in native PAGE, heat denaturation, and RNaseH treatment[27].

**Electrophoretic mobility shift assay**. 5′-End maleimide-IR800-labeled hybrid substrate was incubated with RAD52 or CSB in Buffer B [25 mM Tris-HCl (pH 7.5), 1 mM MgCl$_2$, 1 mM DTT, 50 μg/mL BSA] with 50 mM NaCl for 15 min at 37 °C. Reactions were loaded on 6% PAGE-TBE gel and resolved at 4 °C. Gels were imaged using an Odyssey scanner (LI-COR Biosciences) and quantified.

**Biotin-labeled hybrid pulldown assay**. 3′-Biotin-labeled single ssRNA (oligo 3) and complementary ssDNA (oligo 4) were synthesized from IDT and annealed (Supplementary Table 3). The 293 Flp-in cells overexpressed with GFP-CSB 337–509, 510–960, 961–1493, 1200–1493, and 1400–1493 were lysed in Pierce IP lysis buffer (25 mM Tris-HCl, pH 7.4; 150 mM NaCl; 1 mM EDTA; 1% NP-40; 5% glycerol; and 1 × protease inhibitor cocktail). The pulldown assay was performed using a Pierce magnetic RNA-Protein pulldown kit (Thermofisher Scientific, Catalog#: 20164) following manufacturer's instructions. Briefly, the biotin-labeled hybrids were coated on streptavidin magnetic beads and incubated with cell lysate in buffer (20 mM Tris-HCl, pH 7.5; 50 mM NaCl; 2 mM MgCl$_2$; 0.1% Tween-20; 30% glycerol; 0.2 U/μL RNase), washed in buffer (20 mM Tris-HCl, pH 7.5; 10 mM NaCl; 0.1% Tween-20) and eluted using SDS loading buffer. The binding supernatants and final elutions were analyzed by western blotting.

**Chemical crosslink and MS analysis**. To crosslink the protein complex, the recombinant His-RAD52 and GST-CSB-AD (residues 337–509) were incubated with 1 mM DSS in amine-free buffer (100 mM Hepes, pH 7.5, 50 mM KCl). The cross-linking reactions were performed at 25 ℃ for 1 h with constant agitation (1300 r.p.m.) before subsequent quenching with 50 mM ammonium bicarbonate (final concentration) for 10–15 min. The cross-linked samples were then briefly centrifuged to remove any potential insoluble products and the supernatants were collected for SDS-PAGE analysis.

After protein reduction and alkylation, the cross-linked samples were separated by a 4–12% SDS-PAGE gel (NuPAGE, Thermo Fisher). The regions ( > 250 kDa) corresponding to the cross-linked species were cut and digested with trypsin[36–38]. After proteolysis, the peptide mixtures were desalted and analyzed with a nano-LC 1200 that is coupled online with a QExactive basic mass spectrometer (Thermo Fisher). The detailed peptides were loaded onto a picochip column (C18, 3 μm particle size, 300 Å pore size, 50 μm × 10.5 cm; New Objective) and eluted using an 80 min LC gradient. The QE instrument was operated in the data-dependent mode, where the top six most abundant ions (mass range 350–1,500, charge state > 3) were fragmented by high-energy collisional dissociation (normalized energy 28) and analyzed in the Orbitrap mass analyzer. The target resolution was 75,000 for MS and 15,000 for tandem MS (MS/MS) analyses. The quadrupole isolation window was 1.8 Th and the maximum injection time for MS/MS was set at 800 ms. The data were searched by pLink[39] for identification of cross-linked peptides. The mass accuracies were specified as 10 and 20 p.p.m. for MS and MS/MS, respectively. Other search parameters included cysteine carboxymethylation as a fixed modification and methionine oxidation as a variable modification. A maximum of two trypsin missed-cleavage sites was allowed. The initial search results were obtained using the default 5% false discovery rate, estimated using a target-decoy search strategy. The crosslink spectra were then manually checked to remove potential false-positive identifications from our data set[37,38]. The residue-specific crosslink connectivity map was generated by the online software CX-Circos (www.cx-circos.com).

**STORM microscopy**. The samples for STORM imaging were labeled using the same immunofluorescent staining method described above. S9.6, CSB, and RAD52 were labeled by Alexa Fluor 647; TA-KR was labeled by Cy3B. Alexa Fluor 647-conjugated donkey anti-mouse antibody and Cy3B-conjugated donkey anti-rabbit antibody, which were synthesized in our laboratory (1:5000), were used for two-color STORM imaging[40]. The STORM imaging was performed in our custom-built system on an Olympus IX71 inverted microscope frame with a × 60 oil objective. For data acquisition, continuous illumination with a 642 or 561 nm laser was used in the two-color STORM imaging. The two channels were imaged sequentially at an exposure time of 20 ms for 30,000 imaging frames using 642 nm excitation, followed by 30,000 imaging frames using 561 nm excitation. Fluorescent beads (0.1 μm diameter, F8803, Fisher Scientific, excited using a 488 nm laser) were used as fiduciary markers on the coverslip to correct for 3-dimensional system drift every 200 frames. The reconstruction of the super-resolution image was performed using our custom program written in Matlab 2015 (MathWorks)[41]. Multi-color fluorescence beads (TetraSpeck microspheres, 0.1 μm diameter, blue/green/orange/ dark red fluorescence, Fisher Scientific) were used to correct the chromatic aberration error across different color channels.

## Data availability

The datasets generated during and/or analyzed during the current study are available from the corresponding author upon request. The proteomics data of chemical crosslink and mass spectrometric analysis (CX-MS) analysis has been deposited into MassIVE data repository and the accession code is MSV000082876.

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

## Acknowledgements

This work was supported in part by grants from the National Institutes of Health to L.L. (GM118833) and L.Z. (GM076388 and CA197779), Damon Runyon Cancer Research Foundation (DRG-[2253-16]) to Z.L., and China Scholarship Council to Y.T. and M.D. Funding for the open access charge: National Institutes of Health/GM118833. This project used the UPMC Hillman Cancer Center Cytometry Facility, supported in part by National Institutes of Health P30CA047904.

## Author contributions

Y.T. conducted the major experiments. T.Y., M.D., J.T, Y.X., B.G., J.X., Z.L., S.N., and Y. L. participated in the experiments and data analysis. Y.S., A.S.L., and L.Z. participated in the design of the experiments and edited the manuscript. L.L. provided overall experimental guidance. All authors described their specific contributions and reviewed the manuscript.

## Additional information

**Competing interests:** The authors declare no competing interests.

