## [Peer Review File · Nature Communications]

Reviewers' comments:

Reviewer #1 (Remarks to the Author):

In this manuscript, a previously described, elegant and well-controlled system is used to induce ROS in conjunction with the induction of transcription, to study transcription-coupled repair of oxidative damage at what is called the TA-KR locus. Previously published work with this system (Wei et al, PNAS 2015, ref #4 in this reference list) has shown that the induction of ROS at a site of transcription leads to the recruitment of CSB, Rad51, Rad51C and Rad52 to the TA-KR marked locus and that CSB and active transcription are needed for this recruitment. In that study, the authors also showed that upon RNaseH treatment, the localization of Rad52 to these sites is reduced, implicating R-loops in localization of repair proteins in this pathway, and that the ATPase domain of CSB is needed for Rad52 and Rad51C recruitment. They also previously published that Rad52, Rad51C and CSB interact in a manner that is dependent on transcription (DRB) and that is blocked by RNaseH treatment. They showed as well the persistence of gammaH2AX and 53BP1 foci after knockdown of Rad52, CSB, Rad51C and combinations of the three, suggesting a need for these factors in repair. Finally, they showed that CSB patient lines were sensitive to IR and that this was epistatic with DRB.

This work extends these findings a couple of ways. They show that Rad52 is upstream of Rad51 within this previously defined pathway and that BRCA1 is not required. Note, they did previously publish that BRCA1 and BRCA2 are not recruited in a damage dependent way to this locus. They also show that an acidic domain of CSB can partially rescue varied effects of CSB loss whereas a mutant version of this domain with 22 amino acids converted (E to A) cannot rescue these effects. This is an interesting pathway but unfortunately the previously published work by these authors is very similar and this new study does not move far enough beyond what was previously done to merit publication in Nature Communications. Many of the findings here that are presented as new are already described or are directly inferred from the previously published work, and in fact the authors need to do a much better job of describing previously published work in this manuscript. There are also technical concerns about some of the newer data.

- Figure 1 – the authors use gammaH2AX as a marker of DSBs, but it can be induced by a variety of types of damage and stress. The authors need to establish whether a DSB is inducing this effect or something else. Are DSBs made at this locus? They suggest that this is different from an I-SCE1 induced break, however the I-SCE1 break they induce was made outside of the gene and upstream of the promoter at the TRE array. To really compare effects and conclude ROS has a distinct effect from breaks, the breaks need to be introduced into the transcribed region of an active gene.
- The domain used to rescue CSB loss is the CSB-AD22A mutant. This is a dramatically altered sequence with 22 mutations and it is concerning that this may be very much unfolded. Also a full length version of CSB with the 22A mutations needs to be used for all experiments and not just this domain, particularly given that the authors' previous work showed the CSB ATPase mutation is needed for many of the reported effects (PNAS paper above). What is the impact of the acidic mutations in the context of the ATPase activity of CSB?
- Figure 2C – in this experiment the KO of CSB leads to a reduction of foci positive cells from ~55% to 40% but in Figure 2A, knockdown of CSB leads to a much greater reduction – from about 65-10%. One would expect the knockout to be more effective than the knockdown. What is going on?
- How does a point mutation in the CSB ATPase domain affect the recruitment of Rad51 and repair of ROS-induced damage?
- The authors need to treat their S9.6 IF samples with RNaseA before analysis to remove RNA that

is known to also bind to the S9.6 antibody. They should also use S9.6 drip-sonication or R-CHIP to show at the R-loop forms at this site and not a control site nearby, in an inducible fashion.

- In binding experiments with the RNA-DNA hybrids, the full length CSB is used. Do the CSB-AD domain and the CSB-AD22A mutant also bind?
- Raw foci intensity should be plotted in Figure 4E and 4F for CSB, Rad52 and gammaH2AX. Fold increase may be masking small effects and is not needed. Also, the effect on H2AX-gamma seems extremely mild. There is concern that these effects are not significant.

Minor Comments:

- Figure 3H – why is the CSB-22A protein so much smaller than the wild type and 10A proteins?
- Sup Figure 2B – what is the blot labeled rad51? Is it actually CSB? And why is the knockdown so poor and mostly for the upper band? The KO of CSB removes both bands.
- Rad52 loss causes a G0/G1 pile up – does this matter? Is there any kind of cell cycle effect here?
- Sup Figure 3A – what is the larger band in 337-509? Is it important?
- Previous work from one of these labs showed that RNaseH-D210N stabilized the hybrids formed, but here it does not. Why?

Reviewer #2 (Remarks to the Author):

In this report by Teng et al investigates the mechanism of recombination factor assembly at a defined genome locus as a function of oxidative damage. BRCA1 and BRCA2 are known to be important factors for the loading of the Rad51 recombinase at the site of processed DNA double strand breaks. However, loss of either protein does not result in complete elimination of Rad51 focus formation upon damage, suggesting that additional mechanisms exist for recombination factor recruitment. This work is threaded by the hypothesis that CSB is responsible for recruiting Rad51 to carry out recombinational repair at sites of active transcription. To this end, the authors used a system capable of delivering inducible and localized oxidative damage from the KillerRed protein chromophore at a Tet-controlled chromosomal locus. This intricate system allowed several interesting observations, most important of which are (1) the CSB protein is critical for the recruitment of Rad52 and Rad51 and (2) recruitment of CSB to the site of DNA damage depends on the formation of R loop. The authors went on to show that CSB exhibits strong affinity in vitro toward DNA:RNA hybrid, providing a biochemical basis for the recruitment of CSB to the site of oxidative damage undergoing active transcription. Accordingly, the authors arrived at the main conclusion that CSB is a key factor in assembling recombination repair factors at the site of oxidative damage in a transcription- and R loop-dependent manner. This finding has a high degree of originality and is well-supported by a combination of cell biology, biochemistry, and functional evidence. Overall, the experiments are well-executed and well-controlled. The CSB-RAD52-RAD51 model of ROS damage response has two particular significances in my view. First, it establishes a new paradigm of recombinational repair in the context of transcription and in a BRCA1/2-independent fashion. Second, it identifies a novel of function of CSB in connecting ROS damage-induced R loop formation and the assembly of recombination factors. My critiques are as follows:

1. Fig. 3a-c suggest that the acidic domain per se is capable of localizing as well as recruiting Rad51 to the site of ROS damage. The authors created Glu to Ala substitutions to test the effect of disrupting the CSB-AD domain. Give the number of substitutions made, it is possible that

structural distortions, in addition to charge alterations, are also at play. It would be necessary to test additional CSB-AD mutant to re-inforce the notion that the negatively charged AD region is essential for its enrichment at the site of damage and for the foci formation of Rad51 and Rad52.

2. Fig. 3d, it appears that the CSB 961-1399 fragment is recruited to damaged site more efficiently than both the wild type and the CSB-AD fragment. Is it possible that CSB 961-1399 also contributes to the localization of CSB, although it may not possess the interaction with Rad52? This would be consistent with the result from Fig. 3a.

3. Given that the acidic domain is crucial for CSB function in ROS damage response at the transcribed region, it would be expected that the evolutionary conservation of this domain is high among eukaryotes. Is it the case? The authors should show the sequence alignments of this region to confirm the domain conservation and to justify the design of the Glu to Ala mutants.

Reviewer #3 (Remarks to the Author):

This study investigates the mechanism protecting actively transcribed genome, exploiting the previously established system by the lead author's group, which enables conditional induction of ROS at promoter region with and without transactivation using the KillerRed (KR)-fusions, TA-KR and tetR-KR, respectively. They elegantly demonstrated that 1) transcription-coupled ROS (TA-KR) triggers the accumulation of γ H2AX, which is cleared in a RAD51-dependent manner, 2) RAD51 is recruited at the TA-KR-induced lesion, in a manner dependent on the DNA excision repair protein CSB and the HR factor RAD52, but not on BRCA1/2, 3) the acidic domain (AD) of CSB is responsible for RAD52 recruitment to the TA-KR-induced lesion, 4) the cluster of N-terminal acidic residues in the AD mediates RAD52 interaction, 5) TA-KR induces R-loop formation, which in turn recruits CSB and RAD52, and 6) CSB, as well as RAD52 with lower affinity, directly binds to R-loops.

Together, authors propose that ROS at actively transcribed regions induce R-loops, which in turn recruits CSB, RAD52 and RAD51 in this order. This process is independent of BRCA1/2 and is important to repair DNA damage at these regions.

Experiments are well-executed, and the results are well-presented. The manuscript is consistently well-written and convincing, except one concern regarding their interpretation. It is not clear to this reviewer whether 1) the TA-KR induces double-strand breaks (γ H2AX is a marker of wide-ranging genotoxic stresses, not restricted to double strand breaks), and 2) RAD51 recruited at the TA-KR-induced lesion indeed mediates homologous recombination (HR) repair. It is now well described that R-loops serve as the major source of genotoxic stress, triggering DNA damage when they encounter with DNA replication forks (doi: 10.1016/j.cell.2017.07.043.). Also, RAD51 is shown to catalyse not only the HR repair of broken DNA, but also protect stalled replication forks (doi: 10.1016/j.cell.2011.03.041.). It is conceivable that the ROS-induced lesions and/or R-loops triggers stalled replication forks, which might not be efficiently protected in the absence of the CSB-RAD52-RAD51 axis. This may lead to the persistent γ H2AX, or an increased intensity of γ H2AX (as visible in Fig.1b, γ H2AX at 36 h after ROS induction, comparing siCtrl and siRNA51) after the early repair of ROS-induced lesions by NER. This possibility can be tested by blocking DNA replication (e.g., aphidicolin treatment). If the γ H2AX signal in siRAD51 treated cells is dissolved by DNA replication inhibition, it is very likely that the CSB-RAD52-RAD51 axis protects actively transcribed genome from replicative stress rather than catalysing HR repair.

Reviewer #1:

In this manuscript, a previously described, elegant and well-controlled system is used to induce ROS in conjunction with the induction of transcription, to study transcription-coupled repair of oxidative damage at what is called the TA-KR locus. Previously published work with this system (Wei et al, PNAS 2015, ref #4 in this reference list) has shown that the induction of ROS at a site of transcription leads to the recruitment of CSB, Rad51, Rad51C and Rad52 to the TA-KR marked locus and that CSB and active transcription are needed for this recruitment. In that study, the authors also showed that upon RNaseH treatment, the localization of Rad52 to these sites is reduced, implicating R-loops in localization of repair proteins in this pathway, and that the ATPase domain of CSB is needed for Rad52 and Rad51C recruitment. They also previously published that Rad52, Rad51C and CSB interact in a manner that is dependent on transcription (DRB) and that is blocked by RNaseH treatment. They showed as well the persistence of gammaH2AX and 53BP1 foci after knockdown of Rad52, CSB, Rad51C and combinations of the three, suggesting a need for these factors in repair. Finally, they showed that CSB patient lines were sensitive to IR and that this was epistatic with DRB.

This work extends these findings a couple of ways. They show that Rad52 is upstream of Rad51 within this previously defined pathway and that BRCA1 is not required. Note, they did previously publish that BRCA1 and BRCA2 are not recruited in a damage dependent way to this locus. They also show that an acidic domain of CSB can partially rescue varied effects of CSB loss whereas a mutant version of this domain with 22 amino acids converted (E to A) cannot rescue these effects. This is an interesting pathway but unfortunately the previously published work by these authors is very similar and this new study does not move far enough beyond what was previously done to merit publication in Nature Communications. Many of the findings here that are presented as new are already described or are directly inferred from the previously published work, and in fact the authors need to do a much better job of describing previously published work in this manuscript. There are also technical concerns about some of the newer data.

We thank the reviewer for his/her thoughtful comments. However, we respectfully disagree that the current study is only a modest extension of our previous work. In fact, several of the key findings of our current study are completely unexpected. We highlight some of the novelties of this study below:

1. ***This study reveals that TC-HR is a BRCA-independent pathway for the first time.*** In the previous study, we showed that BRCA1/2 are recruited to sites of ROS induced DNA damage (the reviewer might have misremembered these results). Therefore, it is very surprising that TC-HR is a BRCA1/2-independent pathway. To our knowledge, TC-HR is the first BRCA1/2-independent alternative HR pathway known to date. We believe that this finding represents a significant conceptual advance for the study of HR.

2. ***This study delineates the TC-HR pathway for the first time.*** In the previous study, we found that a number of HR proteins are recruited to sites of ROS induced DNA damage. However, whether and how these proteins function in a pathway was not addressed. In this study, we show that CSB-RAD52-RAD51 function as a linear axis to promote HR, therefore revealing the framework of the TC-HR pathway.

3. ***This study reveals two distinct functional modes of CSB in TC-HR and TC-NER.*** In the previous study, we speculated that CSB functions in TC-HR by binding RNAPII, similar to its function in TC-NER. However, in this study, we found that the TC-HR function of CSB requires the acidic domain, which is dispensable for TC-NER, thereby separating the two distinct functional modes of CSB in TC-HR and TC-NER.

4. This study identifies CSB as a surprising sensor of R loops. Although our previous study suggested that R loops are involved in TC-HR, how R loops function in this pathway was not addressed. In this study, we found that CSB directly binds R loops with a high affinity, suggesting that CSB is a key R-loop sensor to initiate the TC-HR pathway. The identification of CSB as an R-loop sensor will have far-reaching impacts on the research of R loops and the DNA damage response, well beyond the focus of the current study.

Note: New data was summarized in **Rebuttal Figures** as well as main figures.

- Figure 1 – the authors use γ H2AX as a marker of DSBs, but it can be induced by a variety of types of damage and stress. The authors need to establish whether a DSB is inducing this effect or something else. Are DSBs made at this locus? They suggest that this is different from an I-SCE1 induced break, however the I-SCE1 break they induce was made outside of the gene and upstream of the promoter at the TRE array. To really compare effects and conclude ROS has a distinct effect from breaks, the breaks need to be introduced into the transcribed region of an active gene.

We thank the reviewer for these insightful comments. Indeed, ROS can induce multiple types of DNA damage, including DSBs, SSBs, and base oxidations. Among these, DSBs are known to induce γ H2AX robustly. SSBs and base damage do not activate ATM/DNA-PK directly. However, SSBs and base damage may induce DSBs indirectly through collapsing DNA replication forks. To test whether DNA replication contributes to the induction of γ H2AX by TA-KR, we blocked replication with aphidicolin (APH)¹ before activating TA-KR (**Rebuttal Fig. 1A**). Our new result shows that DNA replication is not required for the induction of γ H2AX at the TA-KR site, suggesting that SSBs and base damage are not significant contributors of γ H2AX. Therefore, the γ H2AX at the TA-KR site most likely arises from DSBs directly.

To confirm that γ H2AX is specifically generated at the TA-KR site, we performed CHIP using γ H2AX Ab and PCR primers amplifying a locus at the TA-KR site (**Rebuttal Fig. 1B**). Our result shows that γ H2AX is indeed detected at the TA-KR site by CHIP. In addition, KU70 and KU80, which directly bind to DSBs ends, are recruited to sites of TA-KR within 1 min after KR activation (**Rebuttal Fig. 1C**). These results also support that DSBs are generated at this locus.

We used I-SceI to induce DSBs at an untranscribed locus (upstream of the promoter) and showed that the repair of these DSBs was dependent on BRCA1/2 (original manuscript Sup Fig. 1d). In contrast, ROS-induced DNA damage was repaired in a BRCA1/2-independent, but RAD51- and transcription-dependent manner. The difference in repair may be due to (1) the presence or absence of transcription, and/or (2) a difference between I-SceI and ROS induced DSBs. We have made this clear in the revised manuscript. We would like to point out that the I-SceI induced DSB in the DR-GFP reporter is at a transcribed locus. Previous studies by Maria Jasin and colleagues have demonstrated that BRCA1/2 are important for the repair of this DSB. Therefore, even in transcribed regions, the repair of ROS- and I-SceI induced DSBs may be different. ROS is known to induce SSBs efficiently. SSBs can stall transcription and induce R loops robustly, which may suppress the canonical HR and promote TC-HR (see discussion of our manuscript).

- The domain used to rescue CSB loss is the CSB-AD22A mutant. This is a dramatically altered sequence with 22 mutations and it is concerning that this may be very much unfolded. Also a full length version of CSB with the 22A mutations needs to be used for all experiments and not just this domain, particularly given that the authors' previous work showed the CSB ATPase mutation is needed for many of the reported effects (PNAS paper above). What is the impact of the acidic mutations in the context of the ATPase activity of CSB?

Several lines of evidence suggest that the CSB-AD22A mutant is not an unfolded protein:

- 1) CSB-AD22A is still recruited to the site of TA-KR (Fig. 3h in the original manuscript).

- 2) We found that in addition to RAD52, CSB-AD also interacts with RPA. Importantly, the CSB-AD22A mutant still interacts with RPA (**Rebuttal Fig. 1D**).

Following the reviewer's suggestion, we tested whether the full-length CSB containing the 22A mutations (FL CSB-22A) is able to rescue CSB KO in functional assays. FL CSB-22A did not rescue RAD52 focus formation and γ H2AX resolution in CSB KO cells (**Rebuttal Fig. 1E, 1F**). These results not only confirm our previous findings from CSB-AD22A, but also suggest that the other domains of CSB are not sufficient for TC-HR, highlighting the importance of CSB-AD.

The ATPase activity of CSB is required for both TC-NER and TC-HR. In contrast, AD is not required for TC-NER². Therefore, the ATPase activity of CSB is not dependent on AD. This is consistent with the physical separation of the ATPase and acidic domains in the CSB protein³. CSB may regulate RAD52 through several mechanisms. The data in the current study suggest that CSB binds DNA:RNA hybrids and interacts with RAD52 directly through AD. It is plausible that CSB recruits RAD52 to R loops and/or stimulates its binding to R loops. The ATPase activity of CSB may indirectly facilitate the recruitment of RAD52 by R-loops or its retention by modulating chromatin at sites of DNA damage⁴. Unlike CSB-AD, the CSB ATPase domain alone is not sufficient to partially rescue TC-HR in CSB KO cells. Therefore, it is plausible that CSB-AD is the primary functional domain for TC-HR, whereas the ATPase domain plays a facilitating role in this pathway.

- Figure 2C – in this experiment the KO of CSB leads to a reduction of foci positive cells from ~55% to 40% but in Figure 2A, knockdown of CSB leads to a much greater reduction – from about 65-10%. One would expect the knockout to be more effective than the knockdown. What is going on?

Although KO is expected to deplete CSB protein more completely, the function of CSB may have been partially compensated by a backup pathway during the isolation of KO clones. Given that loss of CSB KO likely increases baseline genomic instability and reduces cell proliferation, compensatory mechanisms are expected to accumulate during clonal selections. We would like to emphasize that we were able to suppress the functional TC-HR defects in both CSB KD and KO cells with wild-type CSB protein, demonstrating that CSB loss is responsible for these defects in both contexts.

- How does a point mutation in the CSB ATPase domain affect the recruitment of Rad51 and repair of ROS-induced damage?

We tested the ability of the CSB ATPase mutant (K538R) to recruit RAD51 and found that it was defective for RAD51 focus formation (**Rebuttal Fig. 1G**). Given the known function of CSB ATPase in chromatin remodeling, it is reasonable to speculate that CSB ATPase promotes RAD52 recruitment by remodeling local chromatin, and indirectly facilitates RAD51 recruitment through RAD52. We focused on CSB-AD in this study because it directly binds RAD52, and because it is required for TC-HR but not TC-NER. The role of CSB ATPase in TC-HR is likely indirect through chromatin remodeling, which is similar to its role in TC-NER.

- The authors need to treat their S9.6 IF samples with RNaseA before analysis to remove RNA that is known to also bind to the S9.6 antibody. They should also use S9.6 drip-sonication or R-CHIP to show at the R-loop forms at this site and not a control site nearby, in an inducible fashion.

We have performed the RNaseA and DRIP experiments suggested by the reviewer. RNaseA treatment indeed reduced the nonspecific S9.6 nuclear staining, but the S9.6 staining at the TA-KR site was not affected (**Rebuttal Fig. 1H**), confirming our original results. Furthermore, our DRIP experiments using two independent primer pairs detected R-loops at the TA-KR locus (**Rebuttal Fig. 1I**). We thank the reviewer for these constructive suggestions.

- In binding experiments with the RNA-DNA hybrids, the full length CSB is used. Do the CSB-AD domain and the CSB-AD22A mutant also bind?

As we pointed out in the original manuscript, CSB localizes to the TA-KR site through a “bipartite” mechanism. Both CSB-AD and a C-terminal fragment of CSB (961-1493) are sufficient to localize to the TA-KR site independently (original manuscript Fig. 3d). Interestingly, the C-terminal fragment of CSB forms foci more efficiently than CSB-AD (original manuscript Fig. 3d), indicating that it may play a more important role in the localization of full-length CSB.

To understand how CSB-AD and the C-terminal fragment are recruited, we tested their binding to RNA-DNA hybrids. CSB-AD (337-509) was captured by RNA-DNA hybrids from cell extracts when highly expressed (**Rebuttal Fig. 2A**). However, purified CSB-AD did not bind RNA-DNA hybrids directly (**Rebuttal Fig. 2B**). These results suggest that CSB-AD binds RNA-DNA hybrids indirectly through another protein. As mentioned above, CSB-AD binds both RAD52 and RPA. Since RPA is a sensor of R loops⁵, CSB-AD could be recruited to R loops by RPA. Indeed, knockdown of RPA diminishes the recruitment of CSB-AD (**Rebuttal Fig. 2C**). The CSB-AD22A mutant is defective for RAD52 binding but retains the ability to bind RPA (**Rebuttal Fig. 1D**), explaining why it is still recruited to the TA-KR site. Together, these results suggest that CSB-AD binds RNA-DNA hybrids indirectly through RPA.

The CSB C-terminal fragments 961-1493 and 1200-1493, but not 1400-1493, were captured by RNA-DNA hybrids from cell extracts (**Rebuttal Fig. 2A**), suggesting that an RNA-DNA binding domain may be present in 1200-1493. Consistently, the fragments 961-1493 and 1200-1493 were able to localize to the TA-KR site (**Rebuttal Fig. 2D**). Importantly, using purified protein, we found that the fragment 1200-1493 bound to RNA-DNA hybrids directly (**Rebuttal Fig. 2E**), thereby revealing the RNA-DNA binding domain of CSB.

Collectively, our results suggest that CSB senses R loops through two mechanisms: it binds RNA-DNA hybrids directly through the C-terminal domain, and it binds the displaced ssDNA in R loops indirectly through RPA. This “bipartite” mechanism may ensure the efficient recognition of R loops by CSB and stabilize the CSB-containing complex at damage sites during TC-HR.

- Raw foci intensity should be plotted in Figure 4E and 4F for CSB, Rad52 and gammaH2AX. Fold increase may be masking small effects and is not needed. Also, the effect on H2AX-gamma seems extremely mild. There is concern that these effects are not significant.

Following the reviewer’s suggestion, we have plotted the raw foci intensity (**Rebuttal Fig. 2F**). The results are similar to those generated using relative foci intensity (foci intensity versus background intensity). We chose to use relative foci intensity in our original analysis because of the background signals of Myc-CSB and GFP-RAD52 caused by their overexpression. Because the baseline level of γ H2AX is high in the cell line, a substantial level of γ H2AX remains even after TA-KR-induced DSBs are repaired. Therefore, even assuming that RNaseH impairs TC-HR, it is only expected to prevent a partial reduction in γ H2AX. Nonetheless, the difference of γ H2AX levels between cells with and without RNaseH is statistically significant (**Rebuttal Fig. 2F**).

Minor Comments:

- Figure 3H – why is the CSB-22A protein so much smaller than the wild type and 10A proteins?

We confirmed that the CSB-AD22A mutant contains the right mutations by sequencing (**Fig. 2G**). As for the reason why it migrates faster than WT and 10A in SDS-PAGE, we speculate that this is due to the charges of these proteins in SDS. Although SDS coats most proteins with negative charges, it has been shown that acidic proteins generally migrate more slowly in SDS-PAGE due to negative charge repulsion with SDS⁶, and that the difference between the predicted and observed MW correlates linearly with the contents of acidic AA⁷. Given that CSB-AD22A is a relative small fragment and it

contains a significantly smaller number of acidic AA than WT and 10A, it is not surprising that it migrates faster than WT and 10A in SDS-PAGE.

- Sup Figure 2B – what is the blot labeled rad51? Is it actually CSB? And why is the knockdown so poor and mostly for the upper band? The KO of CSB removes both bands.

Thanks for pointing out this error. Indeed, it should be labeled as CSB. We have corrected the label.

CSB consistently migrates as two bands in WB. These two bands may represent two isoforms of the CSB protein. It is possible that the CSB siRNA that we used primarily depletes one of the isoforms. The clear effects of CSB siRNA on TC-HR indicate that the residual CSB protein is not functional. For CRISPR KO, we designed an sgRNA to target the ATG start codon, which is expected to eliminate all isoforms of CSB protein. We would like to emphasize that the TC-HR defects of CSB KO and knockdown cells were rescued by wild-type CSB, confirming that the TC-HR defects of these cells are specific.

- Rad52 loss causes a G0/G1 pile up – does this matter? Is there any kind of cell cycle effect here?

In our 2015 PNAS paper⁸, we showed that HR proteins are recruited to the TA-KR site even in G0/G1 cells. Therefore, it is unlikely that RAD52 loss affects TC-HR through cell-cycle changes.

- Sup Figure 3A – what is the larger band in 337-509? Is it important?

This is possibly a nonspecific band or the dimer of AD based on the size.

- Previous work from one of these labs showed that RNaseH-D210N stabilized the hybrids formed, but here it does not. Why?

RNaseH-D210N may preferentially stabilize dynamic or transient R loops. If ROS-generated DNA damage (SSBs and DSBs) induces R loops by stalling RNAPII, these R loops are likely more stable than those formed during unperturbed transcription. Stable R loops may be favorable for the recruitment and retention of DNA repair proteins around DSBs, or allow RNA to play a direct role in DSB repair. This is an interesting question for future studies.

Reviewer #2 (Remarks to the Author):

In this report by Teng et al investigates the mechanism of recombination factor assembly at a defined genome locus as a function of oxidative damage. BRCA1 and BRCA2 are known to be important factors for the loading of the Rad51 recombinase at the site of processed DNA double strand breaks. However, loss of either protein does not result in complete elimination of Rad51 focus formation upon damage, suggesting that additional mechanisms exist for recombination factor recruitment. This work is threaded by the hypothesis that CSB is responsible for recruiting Rad51 to carry out recombinational repair at sites of active transcription.

To this end, the authors used a system capable of delivering inducible and localized oxidative damage from the KillerRed protein chromophore at a Tet-controlled chromosomal locus. This intricate system allowed several interesting observations, most important of which are (1) the CSB protein is critical for the recruitment of Rad52 and Rad51 and (2) recruitment of CSB to the site of DNA damage depends on the formation of R loop. The authors went on to show that CSB exhibits strong affinity in vitro toward DNA:RNA hybrid, providing a biochemical basis for the recruitment of CSB to the site of oxidative damage undergoing active transcription. Accordingly, the authors arrived at the main conclusion that CSB is a key factor in assembling recombination repair factors at the site of oxidative damage in a transcription- and R loop-dependent manner.

This finding has a high degree of originality and is well-supported by a combination of cell biology, biochemistry, and functional evidence. Overall, the experiments are well-executed and well-controlled. The CSB-RAD52-RAD51 model of ROS damage response has two particular significances in my view. First, it establishes a new paradigm of recombinational repair in the context of transcription and in a BRCA1/2-independent fashion. Second, it identifies a novel function of CSB in connecting ROS damage-induced R loop formation and the assembly of recombination factors. My critiques are as follows:

1. Fig. 3a-c suggest that the acidic domain per se is capable of localizing as well as recruiting Rad51 to the site of ROS damage. The authors created Glu to Ala substitutions to test the effect of disrupting the CSB-AD domain. Given the number of substitutions made, it is possible that structural distortions, in addition to charge alterations, are also at play. It would be necessary to test additional CSB-AD mutant to re-inforce the notion that the negatively charged AD region is essential for its enrichment at the site of damage and for the foci formation of Rad51 and Rad52.

We thank the reviewer for these insightful comments. Several lines of evidence suggest that the CSB-AD22A mutant is not an unfolded protein:

- 1) The CSB-AD22A is still recruited to the site of TA-KR (Fig. 3h in the original manuscript).
- 2) We found that in addition to RAD52, the CSB-AD also interacts with RPA. Importantly, the CSB-AD22A mutant still interacts with RPA (**Rebuttal Fig. 1D**).

Following the reviewer's suggestion, we created a CSB-AD12A mutant by changing 12 acidic residues in another region of CSB-AD. We found that the CSB-AD12A mutant still interacts with RAD52 (**Rebuttal Fig. 2G, 2H**). Both AD and AD12A rescued RAD52 foci in CSB KO cells (**Rebuttal Fig. 2I**). These data reinforce the conclusion that the acidic residues mutated in CSB-AD22A are specifically involved in RAD52 binding.

2. Fig. 3d, it appears that the CSB 961-1399 fragment is recruited to the damaged site more efficiently than both the wild type and the CSB-AD fragment. Is it possible that CSB 961-1399 also contributes to the localization of CSB, although it may not possess the interaction with Rad52? This would be consistent with the result from Fig. 3a.

We completely agree with the reviewer's comments. In fact, based on the data mentioned by the reviewer, we suggested that CSB is recruited through a "bipartite" mechanism in the original manuscript.

We have now performed additional experiments to understand how CSB 961-1399 is recruited to the TA-KR site. Using a set of CSB fragments, we found that CSB 961-1493 and 1200-1493, but not 1400-1493, were captured by RNA-DNA hybrids from cell extracts (**Rebuttal Fig. 2A**), suggesting that an RNA-DNA binding domain may be present in 1200-1493. Consistent with these results, CSB 961-1493 and 1200-1493 are sufficient to localize to the TA-KR sites in cells (**Rebuttal Fig. 2D**). Importantly, using purified protein, we found that CSB 1200-1493 binds RNA-DNA hybrids directly (**Rebuttal Fig. 2E**), thereby revealing the RNA-DNA binding domain of CSB biochemically.

We have also investigated how CSB-AD is recruited to the TA-KR site. We found that in addition to RAD52, CSB-AD also binds RPA. Since RPA is a sensor of R loops⁵, CSB-AD may bind R loops indirectly through RPA. Indeed, knockdown of RPA diminished the recruitment of CSB-AD in cells (**Rebuttal Fig. 2C**). The CSB-AD22A mutant is defective for RAD52 binding but retains the ability to bind RPA (**Rebuttal Fig. 1D**), explaining why it is still recruited to the TA-KR site.

Together, our results suggest that CSB senses R loops through two mechanisms: it binds RNA-DNA hybrids directly through the C-terminal domain, and it binds the displaced ssDNA in R loops indirectly through RPA. Our results suggest that CSB uses distinct domains to bind R loops and RAD52, allowing it to bridge R loops and RAD52, and promote the TC-HR function of RAD52 at R loops.

3. Given that the acidic domain is crucial for CSB function in ROS damage response at the transcribed region, it would be expected that the evolutionary conservation of this domain is high among eukaryotes. Is it the case? The authors should show the sequence alignments of this region to confirm the domain conservation and to justify the design of the Glu to Ala mutants.

We thank the reviewer for this interesting question. We compared the sequences of CSB-AD (specifically the 22A region) in human, mouse, zebrafish, and yeast (**Rebuttal Fig. 2J**). Indeed, the 22A region is highly conserved in human and other mammals. In zebrafish, the 22A region is less conserved in sequences but is still enriched with acidic amino acids. This conservation of acidic amino acids justifies the change of Glu to Ala in the CSB-AD22A mutant. In the yeast CSB, we did not observe a sequence conservation with the human CSB 22A region or an enrichment of acidic amino acids. These results suggest that CSB-AD may have been evolved in vertebrates to deal with the ROS-induced genomic instability in complex genomes. It is interesting to note that the function of RAD52 in HR has also evolved during evolution. While the function of RAD52 in the canonical HR pathway is largely taken over by BRCA2 in vertebrates, RAD52 remains important for TC-HR. Perhaps the emergence of CSB-AD is a key event for RAD52 to retain the TC-HR function.

Reviewer #3 (Remarks to the Author):

This study investigates the mechanism protecting actively transcribed genome, exploiting the previously established system by the lead author's group, which enables conditional induction of ROS at promoter region with and without transactivation using the KillerRed (KR)-fusions, TA-KR and tetR-KR, respectively. They elegantly demonstrated that 1) transcription-coupled ROS (TA-KR) triggers the accumulation of γ H2AX, which is cleared in a RAD51-dependent manner, 2) RAD51 is recruited at the TA-KR-induced lesion, in a manner dependent on the DNA excision repair protein CSB and the HR factor RAD52, but not on BRCA1/2, 3) the acidic domain (AD) of CSB is responsible for RAD52 recruitment to the TA-KR-induced lesion, 4) the cluster of N-terminal acidic residues in the AD mediates RAD52 interaction, 5) TA-KR induces R-loop formation, which in turn recruit CSB and RAD52, and 6) CSB, as well as RAD52 with lower affinity, directly binds to R-loops.

Together, authors propose that ROS at actively transcribed regions induce R-loops, which in turn recruits CSB, RAD52 and RAD51 in this order. This process is independent of BRCA1/2 and is important to repair DNA damage at these regions.

Experiments are well-executed, and the results are well-presented. The manuscript is consistently well-written and convincing, except one concern regarding their interpretation. It is not clear to this reviewer whether 1) the TA-KR induces double-strand breaks (γ H2AX is a marker of wide-ranging genotoxic stresses, not restricted to double strand breaks), and 2) RAD51 recruited at the TA-KR-induced lesion indeed mediates homologous recombination (HR) repair. It is now well described that R-loops serve as the major source of genotoxic stress, triggering DNA damage when they encounter with DNA replication forks (doi: 10.1016/j.cell.2017.07.043.). Also, RAD51 is shown to catalyze not only the HR repair of broken DNA, but also protect stalled replication forks (doi: 10.1016/j.cell.2011.03.041.). It is conceivable that the ROS-induced lesions and/or R-loops triggers stalled replication forks, which might not be efficiently protected in the absence of the CSB-RAD52-RAD51 axis. This may lead to the persistent γ H2AX, or an increased intensity of γ H2AX (as visible in Fig.1b, γ H2AX at 36 h after ROS induction, comparing siCtrl and siRNA51) after the early repair of ROS-induced lesions by NER. This possibility can be tested by blocking DNA replication (e.g., aphidicolin treatment). If the γ H2AX signal in siRAD51 treated cells is dissolved by DNA replication inhibition, it is very likely that the CSB-RAD52-RAD51 axis protects actively transcribed genome from replicative stress rather than catalyzing HR repair.

We thank the reviewer for these insightful comments. To examine the possible contributions of replication, we followed the reviewer's suggestion and tested the effects of APH in following experiments (**Rebuttal Fig. 3**).

First, we asked whether DNA replication is required for the formation of γ H2AX and RAD51 foci at the TA-KR site. We used APH to block DNA replication before we induced ROS, and then checked foci formation at an early time point (1 hr) after damage induction. APH treatment did not affect γ H2AX and RAD51 foci (**Rebuttal Fig. 3A**), suggesting that KR induced DSB formation is not dependent on replication. Consistent with this notion, we previously showed that KU80 localizes to the KR site within minutes after damage induction⁸ (**Rebuttal Fig. 1C**). The rapid recruitment of KU80 to this site suggests that DSBs are unlikely generated in a replication-dependent manner.

Second, we asked whether DNA replication is required for the repair of ROS-induced DNA damage. We induced ROS damage and then followed DNA repair in the absence or presence of APH for 24-36 h. As expected, we found that prolonged APH treatments lead to a global increase of γ H2AX (**Rebuttal Fig. 3B upper panel**). This is consistent with the previous observations that APH-induced replication stress leads to accumulation of γ H2AX⁹. In cells transfected with control siRNA, APH did not affect the levels of γ H2AX at the TA-KR locus after 24 h, but slightly reduced γ H2AX levels after 36 h (**Rebuttal Fig. 3B lower panels**). These results show that DNA repair at the TA-KR site occurs even more efficiently in the absence of DNA replication. In cells transfected with RAD51 siRNA, the levels of γ H2AX were higher than those in control cells at 24 and 36 h regardless of the presence or absence of APH (**Rebuttal Fig. 3B lower panels**), confirming that TC-HR remains RAD51-dependent even in the presence of APH. Together, these results suggest that TC-HR is independent of DNA replication.

We are thankful to the reviewer for raising this important question. We believe that our new experiments have significantly improved the main conclusions of this study.

References

1. Hammond, E.M., Green, S.L. & Giaccia, A.J. Comparison of hypoxia-induced replication arrest with hydroxyurea and aphidicolin-induced arrest. *Mutation Research/Fundamental and Molecular Mechanisms of Mutagenesis* **532**, 205-213 (2003).
2. Brosh, R.M., Jr. *et al.* The ATPase domain but not the acidic region of Cockayne syndrome group B gene product is essential for DNA repair. *Molecular biology of the cell* **10**, 3583-3594 (1999).
3. Xu, J. *et al.* Structural basis for the initiation of eukaryotic transcription-coupled DNA repair. *Nature* **551**, 653-657 (2017).
4. Citterio, E. *et al.* ATP-dependent chromatin remodeling by the Cockayne syndrome B DNA repair-transcription-coupling factor. *Molecular and cellular biology* **20**, 7643-7653 (2000).
5. Nguyen, H.D. *et al.* Functions of Replication Protein A as a Sensor of R Loops and a Regulator of RNaseH1. *Molecular cell* **65**, 832-847.e834 (2017).
6. Shirai, A. *et al.* Global analysis of gel mobility of proteins and its use in target identification. *J Biol Chem* **283**, 10745-10752 (2008).
7. Guan, Y. *et al.* An equation to estimate the difference between theoretically predicted and SDS PAGE-displayed molecular weights for an acidic peptide. *Sci Rep* **5**, 13370 (2015).
8. Wei, L. *et al.* DNA damage during the G0/G1 phase triggers RNA-templated, Cockayne syndrome B-dependent homologous recombination. *Proceedings of the National Academy of Sciences* **112**, E3495 (2015).
9. Mazouzi, A. *et al.* A Comprehensive Analysis of the Dynamic Response to Aphidicolin-Mediated Replication Stress Uncovers Targets for ATM and ATMIN. *Cell Rep* **15**, 893-908 (2016).

Rebuttal Figure 1 (For reviewer 1)

Fig. 1 (A) γH2AX foci frequency at TA-KR 1 h after light-induced KillerRed activation in cells treated with or without aphidicolin (APH, 5 $\mu\text{g}/\text{mL}$, 2 h). (B) ChIP assay of U2OS TRE cells transfected with TA-KR after damage was performed using ChIP-IT Express Enzymatic kit (Active Motif) and γH2AX antibody (clone JBW301). A region of TRE locus was amplified by PCR. (C) Recruitment of GFP-tagged Ku70 and Ku80 to TA-KR in U2OS TRE cells. (D) Interactions between Flag-YFP-RPA1 and Myc-CSB-AD WT, 10A and 22A were tested by anti-Myc Co-IP in Flp-in 293 cells. (E) γH2AX foci frequency at TA-KR at early (1 h) and late (36 h) time points after damage in WT, CSB KO cells and CSB KO cells transfected with Myc-CSB WT, 10A or 22A. (F) RAD52 foci frequency at TA-KR in WT, CSB KO cells and CSB KO cells transfected with Myc-CSB WT, 10A or 22A. (G) RAD51 foci frequency at TA-KR in WT, CSB KO cells and CSB KO cells transfected with HA-CSB WT or K538R. (H) DRIP assay detected the fold enrichment of DNA: RNA hybrids at two loci in transcription cassette near TRE region in cells transfected with TA-KR/TA-Cherry ($n=3$). (I) S9.6 foci staining at TA-KR tetR-KR, TA-Cherry and tetR-Cherry in cells treated with RNaseA (100 $\mu\text{g}/\text{mL}$ in RNaseA digestion buffer). The HA-RNaseH1 WT but not D210N overexpression reduced S9.6 foci. For (A), (E) to (G) and (I), $n=3$, 50 cells per replicate. Unpaired t -test, error bars represent SEM. * $P < 0.05$, ** $P < 0.01$, *** $P < 0.001$.

Rebuttal Figure 2 (For reviewer 1 and 2)

(A) Biotin-labeled DNA:RNA hybrid pulldown assay using cell lysates with GFP-CSB 337-509, 510-960, 961-1493, 1200-1493, 1400-1493 expression (B) Purified GST-CSB-AD protein did not show binding to DNA:RNA hybrid in vitro by EMSA assay. (C) Recruitment of GFP-CSB-AD to TA-KR in cells treated with control, RPA1 or RAD52 siRNAs. (D) Recruitments of GFP-CSB 961-1493, 1200-1493 or 1400-1493 to TA-KR were tested and their relative foci intensity on TA-KR was marked on the side (- no foci, ++ strong foci, +++ stronger foci). (E) SDS-PAGE and DNA:RNA hybrid EMSA assay of purified GST-CSB-C ter (1200-1493) protein. Error bars represent SD. (F) Re-plotted figures of CSB, RAD52 and γ H2AX foci using arbitrary intensity (A.U.). (G) Confirmation of CSB-AD WT, 10A, 22A and 12A by sequencing. The amino acids (E, D or A) were marked above the DNA codons. (H) Interactions of Myc-RAD52 with GFP-AD WT, 12A and 22A were tested by anti-GFP Co-IP in Flp-in 293 cells. (I) RAD52 foci frequency at TA-KR in WT, CSB KO cells and CSB KO cells transfected with GFP-AD or GFP-AD12A. (J) Sequence alignment of the CSB-AD 22A region in human (*Homo sapiens*), mouse (*Mus musculus*), zebrafish (*Danio rerio*) and yeast (*Saccharomyces cerevisiae*). Acidic amino acids are marked in purple. For (C), (F) and (I), $n=3$, 50 cells per replicate. Unpaired t -test, error bars represent SEM. * $P < 0.05$, ** $P < 0.01$, *** $P < 0.001$.

Rebuttal Figure 3 (For reviewer 3)

Fig. 3 (A) γH2AX and RAD51 foci frequency at TA-KR 1 h after light-induced KillerRed activation in cells treated with or without aphidicolin (APH, 5 $\mu\text{g/mL}$, 2 h). **(B)** γH2AX foci staining at TA-KR in control or RAD51 knockdown cells treated with or without APH (1 μM , ~0.3 $\mu\text{g/mL}$) for 24 h or 36 h after light-induced KillerRed activation. N=3, 50 cells per replicate. Unpaired *t*-test, error bars represent SEM. **P* < 0.05, ***P* < 0.01, ****P* < 0.001.

Reviewers' comments:

Reviewer #2 (Remarks to the Author):

In this revised manuscript, Teng et al performed several new experiments to enhance the existing data in demonstrating a role for CSB in transcription-dependent recruitment of recombination factors during oxidative DNA damage. These include the involvement of the ATPase activity, a less abrasive mutant of the CSB acidic domain, and additional control data. These new results significantly strengthened the experimental support for the main conclusion. Particularly, the consistent results from the CSB AD12 E-to-A mutant alleviated the concern that structural disruption may be a potential cause of impaired recruitment of recombination factors to the sites of DNA damage.

My remaining suggestion is to include the Rebuttal Figure 2J in which an alignment of the CSB acidic domains from four species is shown. This is informative for the readers to understand the high degree of conserved acidic residues among vertebrates. In this sub figure, "Fish" is presumably zebra fish and should be specified as so.

Reviewer #3 (Remarks to the Author):

The authors essentially addressed my questions, and according to the results they present, I now agree that TA-KR indeed induces double-strand breaks. However, in my view, their observation with Aphidicolin (APH) treated cells following TA-KR-induced ROS damage does not meet with their conclusion that RAD51 catalyses the canonical HR repair at that locus. APH is an inhibitor of B family DNA polymerases, which catalyse not only replicative DNA synthesis but also HR-mediated DNA repair synthesis. A reduced level of gH2AX in APH treated cells, as seen at 36 h time point, somewhat suggests an involvement of another repair mechanism but not HR repair. Or, it is another DNA polymerase (i.e., low fidelity or translesion DNA polymerase), which repairs the DSB? This is an important point conceptually, and has to be fully addressed before the manuscript is accepted for publication.

Response to reviewer 3's comment:

Reviewer #3 (Remarks to the Author):

The authors essentially addressed my questions, and according to the results they present, I now agree that the TA-KR indeed induces double-strand breaks. However, in my view, their observation with Aphidicolin (APH) treated cells following TA-KR-induced ROS damage does not meet with their conclusion that RAD51 catalyses the canonical HR repair at that locus. APH is an inhibitor of B family DNA polymerases, which catalyse not only replicative DNA synthesis but also HR-mediated DNA repair synthesis. A reduced level of gH2AX in APH treated cells, as seen at 36 h time point, somewhat suggests an involvement of another repair mechanism but not HR repair. Or, it is another DNA polymerase (i.e., low fidelity or translesion DNA polymerase), which repairs the DSB? This is an important point conceptually, and has to be fully addressed before the manuscript is accepted for publication.

We thank the reviewer for this thoughtful comment. Indeed, APH is known to inhibit the B-family DNA polymerases, including Pol α and Pol δ ^{1,2}. Notably, the replicative polymerase Pol ϵ is less sensitive to APH than Pol α and Pol δ ³. Therefore, the inhibition of DNA replication by APH in cells is likely a result of Pol α/δ inhibition. A number of DNA polymerases have been implicated in HR by genetic or biochemical studies⁴. These polymerases include members of B-family (Pol α , δ , ϵ , ζ) and Y-family (Pol η , κ , Rev1). For example, Pol η has been shown to localize to sites of DSBs in cells, interact with PALB2 and RAD51, and extend D loops in vitro^{5,6}. It is conceivable that when some of the polymerases in this group are inhibited by APH, others can function redundantly in HR. In our experiment, although chromosomal DNA replication was inhibited by APH, DNA synthesis likely occurred during HR through one or more APH-refractory polymerases.

It is also worth noting the possibility that long-range chromosomal DNA replication may be more sensitive to APH than short-range repair DNA synthesis. Typically, a DNA replication fork has to travel a long distance to collide with an R loop. In the presence of APH, the chance for a fork to collide with an R loop should be drastically reduced. In contrast, the DNA synthesis during repair is generally short. Even if repair DNA synthesis is inefficient in APH, it may occur slowly over time. In our experiment, the repair observed after 36h could also be explained by the differential effects of APH on replication and repair.

We hope that our response has satisfactorily addressed the reviewer's comment. We sincerely thank the reviewer for his/her constructive comments that have helped us improve this manuscript.

- 1 Sheaff, R., Ilsley, D. & Kuchta, R. Mechanism of DNA polymerase alpha inhibition by aphidicolin. *Biochemistry* **30**, 8590-8597 (1991).
- 2 Zhang, Y., Baranovskiy, A. G., Tahirov, T. H. & Pavlov, Y. I. The C-terminal domain of the DNA polymerase catalytic subunit regulates the primase and polymerase activities of the human DNA polymerase alpha-primase complex. *J Biol Chem* **289**, 22021-22034, doi:10.1074/jbc.M114.570333 (2014).
- 3 Syvaaja, J. *et al.* DNA polymerases alpha, delta, and epsilon: three distinct enzymes from HeLa cells. *Proc Natl Acad Sci U S A* **87**, 6664-6668 (1990).
- 4 McVey, M., Khodaverdian, V. Y., Meyer, D., Cerqueira, P. G. & Heyer, W. D. Eukaryotic DNA Polymerases in Homologous Recombination. *Annu Rev Genet* **50**, 393-421, doi:10.1146/annurev-genet-120215-035243 (2016).
- 5 Buisson, R. *et al.* Breast cancer proteins PALB2 and BRCA2 stimulate polymerase eta in recombination-associated DNA synthesis at blocked replication forks. *Cell Rep* **6**, 553-564, doi:10.1016/j.celrep.2014.01.009 (2014).
- 6 McIlwraith, M. J. *et al.* Human DNA polymerase eta promotes DNA synthesis from strand invasion intermediates of homologous recombination. *Mol Cell* **20**, 783-792, doi:10.1016/j.molcel.2005.10.001 (2005).

REVIEWERS' COMMENTS:

Reviewer #3 (Remarks to the Author):

Authors addressed my question satisfactorily.